# A synthetic BRET-based optogenetic device for pulsatile transgene expression enabling glucose homeostasis in mice

Ting Li[1,2,5], Xianjun Chen [1,2,3,5], Yajie Qian[1,2], Jiawei Shao[4], Xie Li[1,2], Shuning Liu[1,2], Linyong Zhu [1], Yuzheng Zhao [1,2], Haifeng Ye [4✉] & Yi Yang [1,2,3✉]

Pulsing cellular dynamics in genetic circuits have been shown to provide critical capabilities to cells in stress response, signaling and development. Despite the fascinating discoveries made in the past few years, the mechanisms and functional capabilities of most pulsing systems remain unclear, and one of the critical challenges is the lack of a technology that allows pulsatile regulation of transgene expression both in vitro and in vivo. Here, we describe the development of a synthetic BRET-based transgene expression (LuminON) system based on a luminescent transcription factor, termed luminGAVPO, by fusing NanoLuc luciferase to the light-switchable transcription factor GAVPO. luminGAVPO allows pulsatile and quantitative activation of transgene expression via both chemogenetic and optogenetic approaches in mammalian cells and mice. Both the pulse amplitude and duration of transgene expression are highly tunable via adjustment of the amount of furimazine. We further demonstrated LuminON-mediated blood-glucose homeostasis in type 1 diabetic mice. We believe that the BRET-based LuminON system with the pulsatile dynamics of transgene expression provides a highly sensitive tool for precise manipulation in biological systems that has strong potential for application in diverse basic biological studies and gene- and cell-based precision therapies in the future.

[1] Optogenetics & Synthetic Biology Interdisciplinary Research Center, State Key Laboratory of Bioreactor Engineering, Shanghai Collaborative Innovation Center for Biomanufacturing Technology, East China University of Science and Technology, 130 Mei Long Road, Shanghai 200237, China. [2] School of Pharmacy, East China University of Science and Technology, 130 Mei Long Road, Shanghai 200237, China. [3] CAS Center for Excellence in Brain Science and Intelligence Technology, Institute of Neuroscience, Chinese Academy of Sciences, Shanghai 200031, China. [4] Synthetic Biology and Biomedical Engineering Laboratory, Biomedical Synthetic Biology Research Center, Shanghai Key Laboratory of Regulatory Biology, Institute of Biomedical Sciences and School of Life Sciences, East China Normal University, Dongchuan Road 500, Shanghai 200241, China. [5] These authors contributed equally: Ting Li, Xianjun Chen. ✉email: hfye@bio.ecnu.edu.cn; yiyang@ecust.edu.cn

Being able to analyze gene expression patterns is essential for understanding protein function, biological signaling pathways and cellular responses to external and internal stimuli. Conventionally, scientists assume that under constant external conditions, the concentrations and activities of key cellular regulatory molecules generally remain steady over time or fluctuate stochastically around fixed mean values[1]. However, recent studies have shown that many regulatory proteins, including transcription factors (TFs), are regulated in a pulsatile fashion, leading to a variety of phenotypic consequences, ranging from stress response to differentiation[1–3]. Despite the fascinating discoveries made in the past few years, it remains unclear what functional capabilities such pulsatile dynamic regulation can provide for cells[1,2]. In addition, the mechanisms of these pulsing systems have been elucidated in a very limited number of cases[4–7], revealing both shared features and qualitative differences across systems[1]. Given the importance of the pulsing systems, developing regulatable transgene systems to precisely control the dynamics of key signaling proteins would be useful for understanding the mechanisms and complex biological functions underlying different pulsing systems[1,3].

Previously, chemically regulated gene expression systems have been widely used for the temporal control of gene expression in numerous biological studies[8–10]. However, gene expression from these systems continues for quite a long time once the inducers are presented, as it takes time for these chemical inducers to be degraded or metabolized[11]. Although gene expression could be terminated by physical removal of the inducers by replacement with fresh medium in vitro, it is almost impossible to dynamically regulate transgene expression in animals simply by controlling the presence or absence of chemical inducers[11]. Thus, these systems are not applicable for controlling the dynamics of regulators in most pulsing systems. To overcome these limitations, a series of light-inducible systems have been developed that allow control of gene expression with unprecedented spatial and temporal precision[12,13]. These optogenetic systems provide robust and convenient tools to achieve pulsatile dynamic regulation of regulators, allowing direct comparison between functional outcomes and concentration-time relationships of external stimuli. For example, the LightOn system developed by our group was used to dynamically control the expression of the transcription factor Ascl1, a Hes1 target that oscillates in neural progenitors, revealing that the dynamics of Ascl1 expression can influence cell fate[14]. Nevertheless, supplying adequate light intensity to activate these optogenetic systems is a major obstacle for their in vivo application, as transmission of external light through deep tissues is extremely inefficient due to light scatter and tissue absorption. Although surgically implanted optical fibers can be used to efficiently deliver external light, undesirable effects (e.g., photobleaching, phototoxicity) may compromise cell viability and experimental outcomes, thereby imposing significant limitations for potential clinical applications[15].

To bypass the challenges associated with the delivery of external light, neuroscientists have developed luminescent opsins by directly coupling bioluminescent proteins to conventional light-sensitive opsins, allowing interrogation of neuronal circuits at different temporal and spatial resolutions by choosing either extrinsic physical or intrinsic biological light for their activation[15,16]. Unlike traditional chemical inducers, luciferase substrates could be readily consumed to produce bioluminescence to activate opsins via bioluminescence resonance energy transfer (BRET). When the substrates were consumed, the activated photosensitive modules gradually recovered to the inactive states, leading to inactivation of opsins[15,16]. Thus, these luminescent opsins allowed regulation of neuron activity in a pulsatile fashion[15,16]. We therefore hypothesized that similar concepts could be used to develop a BRET-based

transgene expression system that allows pulsatile regulation of gene expression both in vitro and in vivo.

Here, we describe the development of a luminescent transcription factor termed luminGAVPO that allows pulsatile and quantitative activation of transgene expression in both mammalian cells and animals in a furimazine- or light irradiance-dependent manner. We further demonstrate the usefulness of the luminGAVPO-based transgene expression system for controlling blood glucose homeostasis in type 1 diabetic mice.

## Results

**Design and construction of the LuminON transgene expression system.** The light-oxygen-voltage (LOV) domain family of proteins is widespread in biology, imparting sensory responses to signal transduction domains[17]. Dark-state LOV proteins contain an oxidized flavin cofactor that maximally absorbs blue light ($\lambda = 440–480$ nm). Upon blue light absorption, a covalent adduct is formed between the C4a position of the noncovalently bound flavin cofactor and the thiol moiety of the active-site cysteine[18–23]. Many of these small and light-responsive LOV modules have been used for the construction of several optogenetic systems[17,24–31]. NanoLuc (Nluc), an engineered luciferase derived from a deep sea shrimp, is a small (19 kDa) yet stable luciferase that produces 100-150× brighter and more sustained luminescence than the traditional firefly (Fluc) or Renilla (Rluc) luciferase[32]. In particular, Nluc utilizes its substrate to produce blue light (spectral maximum 454 nm)[32], which matches the absorption spectra of LOV proteins. Taking advantage of these attributes, we sought to utilize Nluc to serve as the luminescence donor to activate its linked LOV-based optogenetic modules, thereby modulating their activities. To this end, we first attempted to use Nluc to activate the LOVTRAP, an optogenetic approach capable of repeatedly and reversibly controlling protein dissociation using the small protein Zdk that binds to the dark state of LOV2[33]. We fused Nluc to the N terminus of the mCherry-LOV2 fusion protein (Supplementary Fig. 1a). When we coexpressed the protein with a mitochondria-anchored mVenus-Zdk protein, both mCherry and mVenus fluorescence accumulated at mitochondria under dark conditions (Supplementary Fig. 1b and c); however, the mCherry fluorescence readily diffused to the cytosol within 5 s after the addition of furimazine or exposure to external blue light (Supplementary Fig. 1b–e); when furimazine was consumed or light illumination ceased, mCherry fluorescence accumulated at mitochondria again (Supplementary Fig. 1d and e). Such translocation of mCherry could be triggered repetitively to the same magnitude with little or no loss in efficacy (Supplementary Fig. 1e). In addition to the LOV-based optogenetic modules, Nluc could also be used to control the recruitment of the CRY2-mCherry-Nluc fusion protein to the plasma membrane[24], where it localized with CIBN-pmEGFP (Supplementary Fig. 1f–i), by activating the interaction between CRY2 and CIB1. These data indicate that Nluc can serve as the luminescence donor for activation of optogenetic modules with matched spectra with Nluc emission.

We previously developed a series of blue light-inducible transgene expression systems based on Vivid (VVD), the smallest LOV domain-containing protein[20,27,30,34,35]. In these systems, the light-switchable TFs dimerize and bind to the promoter region to directly repress gene transcription[36] or activate gene transcription by a fused transcriptional regulatory domain upon blue light exposure[27,34,37]. We next aimed to create a luminescent transcription factor by engineering a chimeric protein that consists of Nluc and a light-switchable transcription factor, GAVPO[27] (Fig. 1a, Supplementary Fig. 2). We constructed different configurations of Nluc-GAVPO fusion proteins by inserting Nluc into the N terminus of the Gal4 domain (NG), C terminus of the VVD

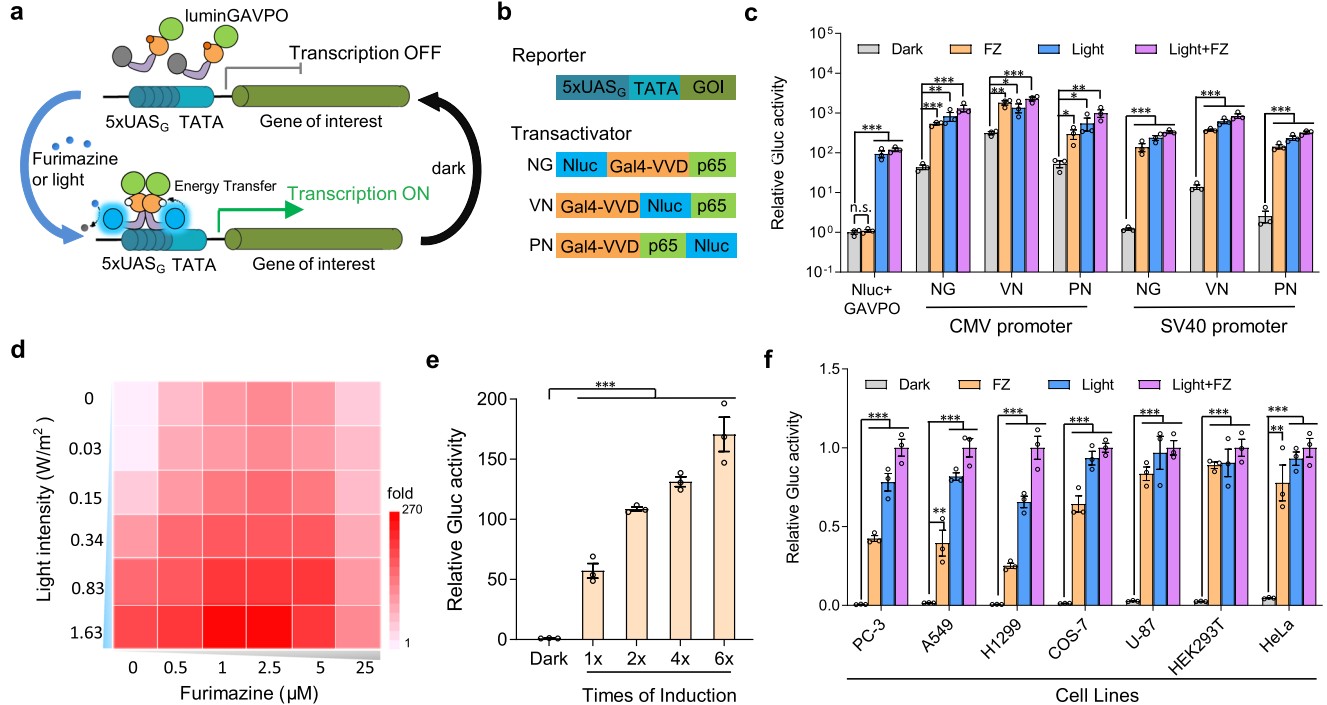

**Fig. 1 Design and construction of the LuminON transgene expression system. a** Schematic representation of the LuminON system. In the presence of furimazine or blue light, luminGAVPO homodimerizes, interacts with $UAS_G$ elements ($5xUAS_G$) and initiates transgene expression of the gene of interest (GOI). **b** Schematic representation of the reporter plasmid containing $5xUAS_G$ upstream of the TATA box and the transactivator plasmids encoding different configurations of Nluc-GAVPO fusion proteins. **c** Furimazine (FZ)- or blue light-induced Gluc expression by different configurations of Nluc-GAVPO transactivators driven by the CMV or SV40 promoter. **d** Quantitative control of Gluc expression by modulating furimazine concentration and blue light intensity. The transfected cells were cultured with different furimazine concentrations (0–25 µM) or light irradiances (0–1.63 W/m²). Gluc activity was determined 24 h after induction, and induction ratios were calculated. **e** Gluc expression kinetics upon induction by furimazine for different durations. The transfected cells were induced by furimazine for different durations within 12 h, and Gluc activity was measured 24 h after the first induction. Data were normalized to Gluc expression in the cells kept in dark conditions without induction by furimazine. **f** LuminON system-mediated Gluc expression in different mammalian cell types. Data were normalized to Gluc expression simultaneously induced by furimazine and external blue light. Data in (**c**, **e** and **f**) represent the mean ± s.d. from three technical replicates. All the statistical comparison was performed by two-tailed $t$ test. n.s. means not significantly different, $*P < 0.05$, $**P < 0.01$, $***P < 0.001$. Source data are provided in the Source Data file.

domain (VN) and p65 domain (PN) in GAVPO (Fig. 1b). We tested their furimazine- or light-dependent impact on the transcriptional activity of a Gaussia luciferase (Gluc) reporter driven by Gal4 binding sites upstream of a TATA box after transient transfection into HEK293 cells (Supplementary Note 1). These results showed that all the Nluc-GAVPO fusions could significantly activate Gluc expression upon incubation with furimazine (FZ) or illumination with external blue light (Fig. 1c), whereas coexpression of Nluc and GAVPO separately yielded only a response to external blue light (Fig. 1c), indicating that furimazine-induced Gluc expression depends on the close proximity of Nluc and VVD in GAVPO to allow BRET to occur. Further studies showed that the interference of luminGAVPO in the cell supernatants was minimal (Supplementary Fig. 3a). We also noticed that Nluc-GAVPO fusions driven by a strong CMV promoter had higher activation levels than those driven by a weak SV40 promoter, but with much higher background expression in the uninduced states (Fig. 1c). It seemed that high Nluc-GAVPO expression resulted in homodimer formation independent of furimazine or light illumination (Supplementary Fig. 4a), thereby activating promoter activities even under dark conditions, which was further validated by transfecting lower amounts of the CMV-driven NG expression plasmid into cells, resulting in lower background expression and higher induction ratios (Supplementary Fig. 4b and c). Since Nluc-GAVPO with NG configuration had moderate activation levels yet exhibited maximal induction ratio by

furimazine, we termed it luminGAVPO (luminescent GAVPO), and the transgene expression system based on luminGAVPO was named the LuminON system.

As luminGAVPO could be activated by either furimazine or external blue light, a digital "OR"-like logic gate was predicted for luminGAVPO-activated transgene expression using furimazine and external blue light as the two inputs. We observed increased Gluc expression along with increasing furimazine or blue light irradiance (Fig. 1d, Supplementary Fig. 3b and c). Such dual inputs also support the development of conceptually complex applications. Notably, although higher concentrations of furimazine could produce brighter and more sustained luminescence (Supplementary Fig. 5a and b), Gluc expression slightly decreased when the furimazine concentration was higher than 5 µM (Fig. 1d, Supplementary Fig. 3b). The maximum induction ratio obtained by induction with furimazine alone could reach ~116-fold when the furimazine concentration was 2.5 µM (Supplementary Fig. 3b), and the induced mRNA could be directly visualized using the Pepper fluorescent RNA recently developed by our group[36] (Supplementary Fig. 6). In addition, highly precise control of transgene expression was achieved by combining the furimazine concentration and light irradiance (Fig. 1d). Notably, furimazine exhibited slight inhibition of protein synthesis at high concentrations with more than 5 µM (Supplementary Fig. 7a), but little effects on cell proliferation and viability were observed (Supplementary Fig. 7b).

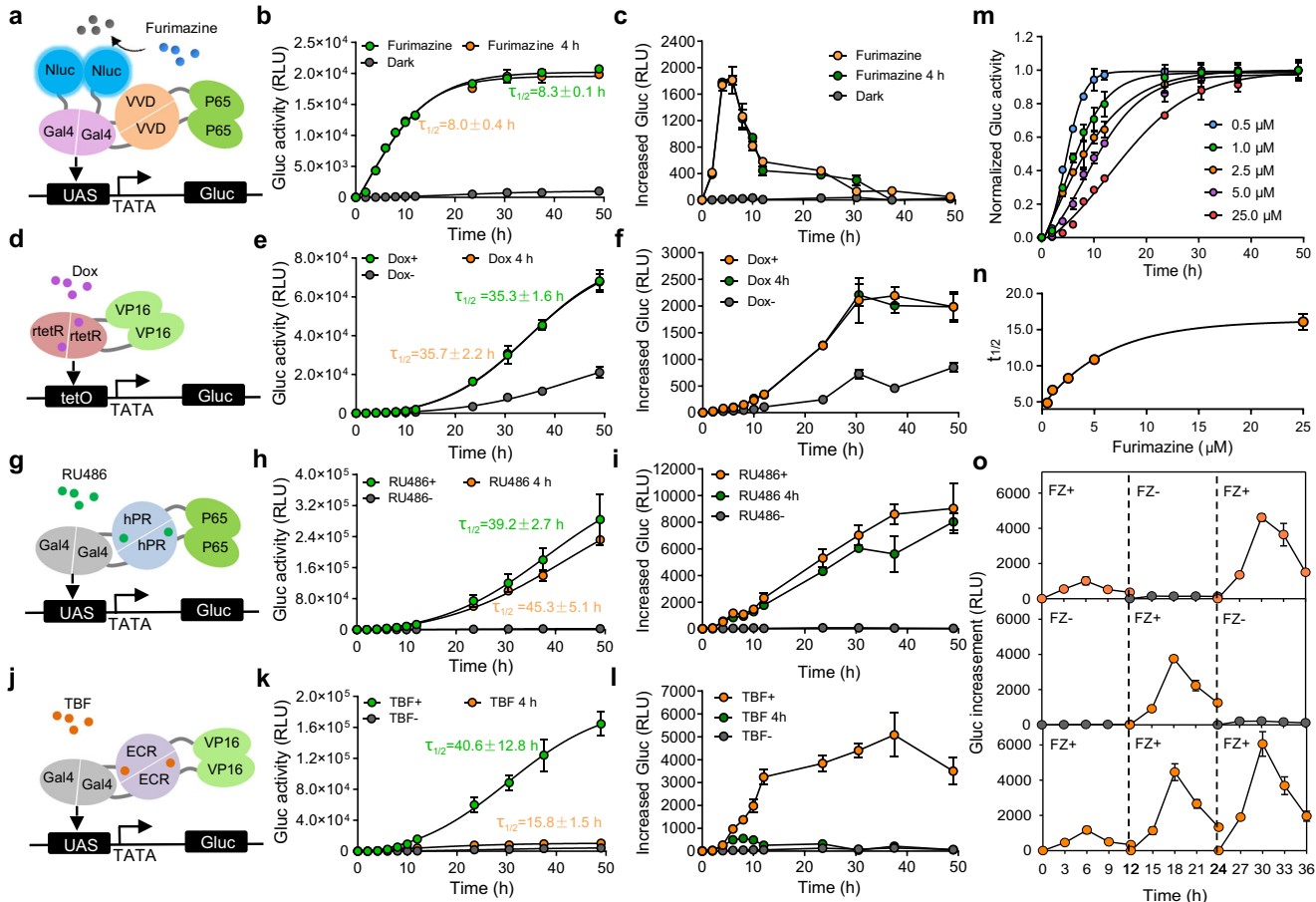

**Fig. 2 Pulsatile activation of transgene expression by luminGAVPO.** Time course of Gluc expression kinetics mediated by different chemical-inducible transgene expression systems induced by furimazine (**a–c**), Doxycyline (Dox) (**d–f**), Mifepristone (RU486) (**g–i**) or Tebufenozide (TBF) (**j–l**). HEK293 cells were transfected with plasmids encoding the corresponding components of different systems. The cells were induced by 2.5 µM furimazine, 1 µM Dox, 1 µM TBF or 0.1 µM RU486 for 24 h or only for 4 h. Gluc activity at the indicated time points after induction was measured, and $\tau_{1/2}$ values are indicated (**b**, **e**, **h**, **k**). The increase in Gluc expression between a time point and the previous time point is replotted from (**b**, **e**, **h**, **k**) and is indicated as (**c**, **f**, **i**, **l**), respectively. Cells kept in the dark without induction were used as controls. **m** Time course of Gluc expression kinetics induced by different concentrations of furimazine. Data were normalized to the maximal Gluc expression. **n** $\tau_{1/2}$ values of Gluc expression induced by different concentrations of furimazine. **o** Reversible activation of Gluc expression mediated by luminGAVPO. The transfected cells were treated with different programmed durations of furimazine induction within 36 h. The medium was refreshed every 12 h to remove the accumulated Gluc. Data represent the increase in Gluc expression between a time point and the previous time point. Data in (**b–c**, **e–f**, **h–i** and **k–o**) represent the mean ± s.d. from three technical replicates. Source data are provided in the Source Data file.

While the luminescence exhibited a flash-type profile and could only be sustained for 2–3 h for 2.5–5 µM furimazine (Supplementary Fig. 5), we wondered whether induction by furimazine multiple times could increase the luminGAVPO-mediated activation levels, considering that Nluc is resistant to autoinhibition by its catalytic byproducts[32]. To this end, we induced the cells with multiple rounds of furimazine within 12 h (Supplementary Fig. 8), and Gluc expression was measured 24 h after the first induction. The results showed that Gluc expression increased with increasing round of induction by furimazine (Fig. 1e). Furthermore, the LuminON system exhibited excellent transgene expression properties in various mammalian cell lines, indicating its broad application potential (Fig. 1f).

**Pulsatile activation of transgene expression mediated by luminGAVPO.** As shown previously, Nluc catalyzes furimazine to display a relatively short-lived, flash-type light emission profile (Supplementary Fig. 5). When furimazine is consumed, the activated luminGAVPO is gradually inactivated and dissociates

from its cognate DNA binding site, leading to inactivation of transcription (Fig. 2a). Thus, furimazine-induced transgene expression by luminGAVPO would display a pulsatile profile. To validate our hypothesis, we measured the time course of Gluc expression upon furimazine induction. Our results showed that gene transcription and expression occurred rapidly, as shown by the sharp increase in Gluc mRNA levels and protein production in the cells within 2 h after furimazine induction (Fig. 2b, Supplementary Fig. 9). Thereafter, we observed a fast decay in the amount of Gluc mRNA and a plateau in the amount of protein without changing the medium (Fig. 2b, Supplementary Fig. 9), which is consistent with the luminescence profile of Nluc-catalyzed furimazine. Thus, furimazine could induce pulsatile activation of transgene expression, with a $\tau_{1/2}$ (the time to reach 50% of maximal expression) of 8.3 ± 0.1 h (Fig. 2a and b). This pulsatile activation profile could be more clearly shown by the increased Gluc production during the indicated time periods, which reflected the rate of Gluc synthesis over time, showing that the Gluc synthesis rate reached a maximum at ~4–6 h after furimazine induction and then sharply decreased (Fig. 2c). Notably,

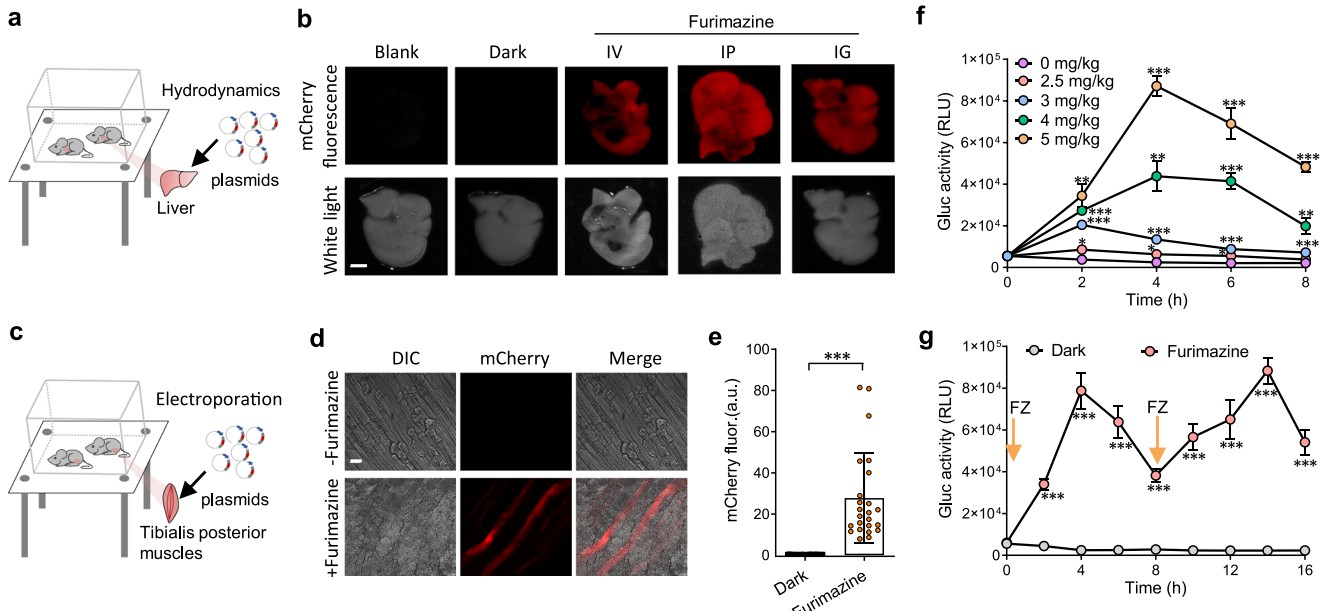

**Fig. 3 Pulsatile activation of transgene expression mediated by luminGAVPO in mice. a** Schematic representation of luminGAVPO-activated transgene expression in mouse livers. The KM mice were transiently hydrodynamically injected (tail vein) with an iteration of the luminGAVPO system comprising the transactivator and reporter plasmids. **b** The transfected mice in (**a**) were kept in the dark and treated with 5 mg/kg furimazine via tail intravenous (IV), intraperitoneal (IP) or intragastric (IG) injection. The mice were sacrificed, and the livers were dissected for mCherry fluorescence imaging. Scale bar, 0.5 cm. **c** Schematic representation of luminGAVPO-activated transgene expression in mouse tibialis posterior muscles. The luminGAVPO and reporter plasmids were transfected into the tibialis posterior muscles via electroporation. **d** The transfected mice in (**c**) were kept in the dark and treated with or without 5 mg/kg furimazine via IV administration. Twenty-four hours after induction, the mice were sacrificed, and the skeletal muscles were dissected, sectioned and imaged. Scale bar, 100 μm. **e** Quantification of mCherry fluorescence in muscle cells. Statistical comparison was performed by a two-tailed $t$ test. ***$P < 0.001$ versus control. Data represent the mean ± s.d. ($n = 24$ cells from 3 mice). **f** Time course of Gluc expression kinetics induced by different concentrations of furimazine. **g** Reversible activation of Gluc expression induced by furimazine. The transfected mice were kept in the dark and treated with 5 mg/kg furimazine via IV administration at 6 h and 14 h after plasmid transfection, respectively. Gluc activity in the bloodstream was measured at the indicated time points after the first induction. All statistical comparisons were performed by a two-tailed $t$ test. *$P < 0.05$, **$P < 0.01$, ***$P < 0.001$ versus control. Data in (**b**, **d**, **f** and **g**) represent the mean ± s.d. from three (**b**, **d**, **f**) or five (**g**) mice. For (**b** and **d**), at least two independent experiments were carried out with similar results. Source data are provided in the Source Data file.

changing the medium to remove the residual furimazine at 4 h after induction did not significantly alter the kinetics of Gluc expression (Fig. 2b and c), probably because the majority of the furimazine had been consumed within 4 h after induction. In contrast, the commonly used chemical-inducible systems maintained high Gluc synthesis rates and showed a rapid increase in Gluc accumulation even 50 h after induction, with a significantly prolonged $\tau_{1/2}$ (>30 h) (Fig. 2d–l). One exception was the EcR system, in which Gluc production dramatically slowed down after removal of Tebufenozide (TBF) but still with a long $\tau_{1/2}$ of 15.8 ± 1.5 h (Fig. 2k and l). Nevertheless, such reversibility can hardly be applied for in vivo studies. Notably, such in vitro comparison does not necessarily translate to the in vivo situation, as the half-life of various chemicals may differ vastly.

Notably, both the pulse amplitude and duration of transgene expression were highly dependent on the dose of furimazine. Lower doses of furimazine exhibited shorter durations (shortened $\tau_{1/2}$) and smaller amplitudes of pulsatile activation, while higher doses of furimazine had longer durations (prolonged $\tau_{1/2}$) and larger amplitudes (Fig. 2m and n). Such tunability will be especially useful for studying pulsing systems with different pulsing dynamics. Further studies showed that pulsatile activation mediated by luminGAVPO could be repetitively and reversibly switched on by alternating furimazine induction at 12-h intervals (Fig. 2o, Supplementary Fig. 10).

**Pulsatile activation of transgene expression in mice.** To determine whether luminGAVPO could be used to activate transgene expression in mice, we transfected luminGAVPO and mCherry reporter plasmids into the livers of mice using a hydrodynamic procedure[38] (Fig. 3a). Marked mCherry fluorescence was observed in the livers of the mice administered furimazine via tail intravenous (IV), intraperitoneal (IP) or intragastric (IG) injection (Fig. 3b), whereas the livers of the mice kept in the dark showed minimal mCherry fluorescence (Fig. 3b). Similar results were observed in the tibialis posterior muscles transfected with the same plasmids by electroporation and induced by furimazine via IV administration (Fig. 3c–e).

After confirmation of the functionality of luminGAVPO in mice, we next measured the time course of furimazine-induced transgene expression in mice transfected with luminGAVPO and pU5-Gluc plasmids. Gluc expression rapidly occurred and reached a maximum level within 4 h after furimazine induction (Fig. 3f). Unlike reaching a plateau in cultured cells, the Gluc protein level in the bloodstream sharply decreased after reaching a maximum level (Fig. 3f) because the half-life of Gluc in the bloodstream (20 min) is significantly shorter than that in culture medium (6 days)[39]. In addition, a lower dose of furimazine exhibited shorter durations and smaller amplitudes of Gluc expression in the bloodstream, consistent with the results from the cultured cells. Moreover, the pulsatile activation profile mediated by luminGAVPO could be repetitively and reversibly induced by administration of furimazine at 8-h intervals (Fig. 3g). Collectively, these results demonstrate that the LuminON system can be used for pulsatile and reversible activation of transgene expression in vivo.

**Pulsatile expression of insulin to enhance blood glucose homeostasis in T1D mice.** Insulin is a peptide hormone secreted by the β cells of the pancreas during nutrient uptake and maintains glucose homeostasis by inhibiting hepatic glucose output while facilitating cellular glucose uptake[40]. This pulsatile secretion of insulin may match the expression profile mediated by luminGAVPO. As a proof-of-concept demonstration of the potential application of the LuminON system for controllable cell therapy, we constructed HEK_FUR-Gluc-P2A-mINS cells stably engineered for luminGAVPO-mediated insulin production. To this end, luminGAVPO and UAS_G-TATA-Gluc-P2A-mINS expression cassettes were integrated into HEK293T cells using the Sleeping Beauty transposon system[41] (for a detailed procedure, see the Online method). Hundreds of clones were picked and validated by detecting Gluc expression with or without induction by furimazine (Supplementary Fig. 11a). Several clones manifesting >30-fold activation of Gluc expression were further validated

(Supplementary Fig. 11b and c). The I18 clone that showed high activation level with low background was chosen for further studies (Supplementary Fig. 11b and c). I18 cells were microencapsulated into coherent, semipermeable and immunoprotective alginate-poly-(L-lysine)-alginate beads that allow free diffusion of substances with low molecular weights (<72 kDa) while shielding their cellular content from physical contact with the host's immune system[42–45] (Fig. 4a). T1D mice receiving the microencapsulated cells were treated with 5 mg/kg furimazine, and their blood glucose levels were measured after 3 h. T1D mice implanted with HEK_FUR-Gluc-P2A-mINS cells exhibited significant restoration of blood glucose levels (Fig. 4b).

In addition, a higher dose of furimazine resulted in more significant restoration of blood glucose and prolonged the maintenance of glucose homeostasis (Fig. 4c). Furthermore, the implanted T1D mice also showed significantly improved glucose tolerance upon furimazine administration (Fig. 4d). Notably, the

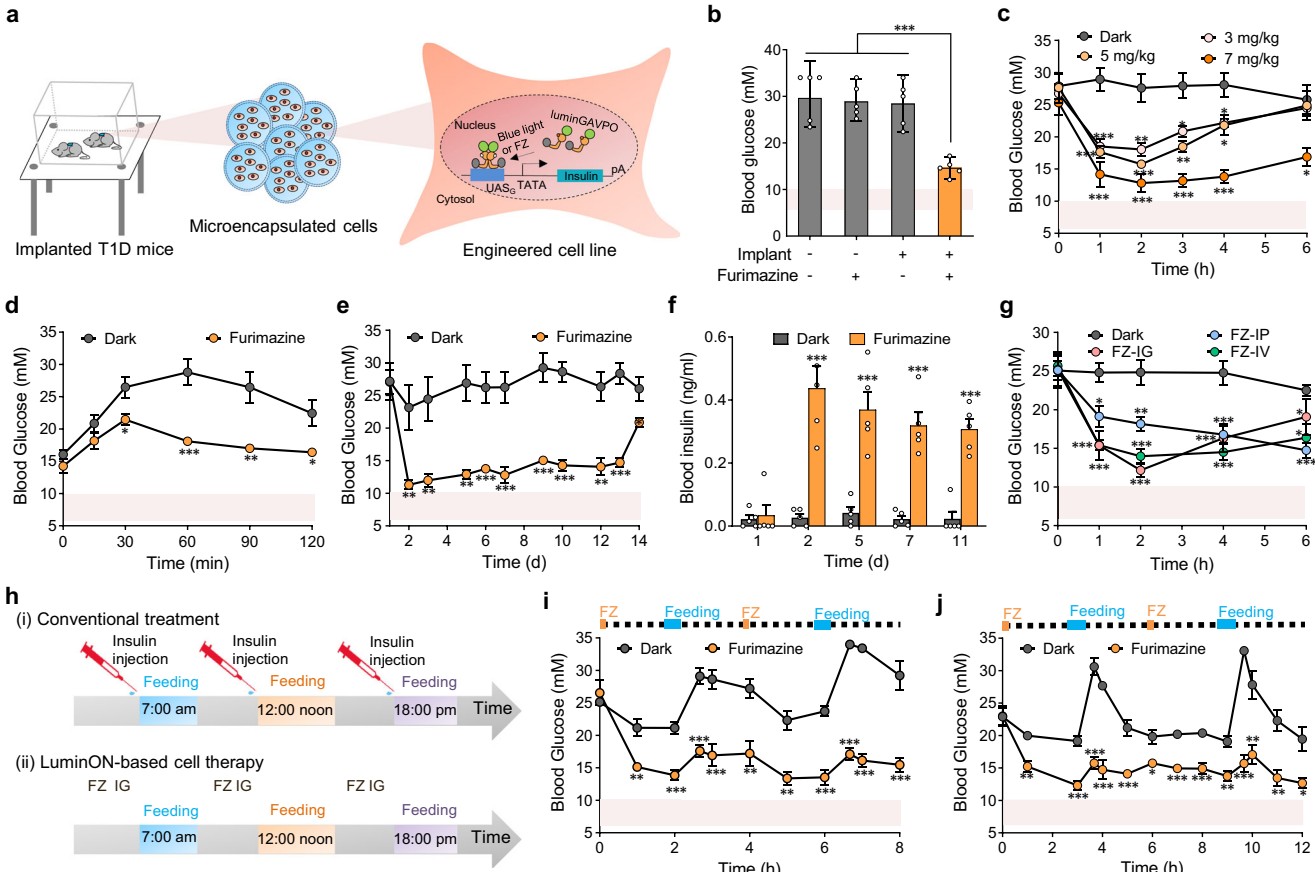

**Fig. 4 Pulsatile expression of insulin to enhance blood-glucose homeostasis in type 1 diabetic (T1D) mice. a** Schematic representation showing luminGAVPO-mediated insulin expression to enhance blood glucose homeostasis in T1D mice. HEK_FUR-Gluc-P2A-mINS cells stably engineered for luminGAVPO-mediated insulin production were microencapsulated into alginate-poly-(L-lysine)-alginate beads and intraperitoneally implanted into T1D mice. **b** Restoration of blood glucose in T1D mice induced by furimazine. Mice implanted with microencapsulated 4 × 10⁶ HEK_FUR-Gluc-P2A-mINS cells were treated with 5 mg/kg furimazine via IV administration, and blood glucose levels were measured 3 h after induction. T1D mice without implantation or without furimazine administration were used as controls. **c** Blood glucose levels in the implanted mice upon administration of different dose of furimazine via IV. **d** Intraperitoneal glucose tolerance test (IGTT) of the implanted T1D mice. The implanted mice received an intraperitoneal injection of aqueous 1 g/kg D-glucose, and the glucose levels of the mice were profiled from 1 to 120 min. **e, f** Long-term maintenance of blood-glucose homeostasis in T1D mice. From day 2, the implanted mice were intragastrically injected with 7 mg/kg furimazine each day, and blood glucose levels (**e**) and insulin production (**f**) were profiled for up to 13 days after implantation. **g** Blood glucose levels in the implanted mice upon administration of furimazine via different delivery methods (IG, IP, or IV). **h** Schematic showing the time schedule and experimental procedure of conventional therapy and LuminON-based cell therapy for combating diabetes. The implanted mice were treated with (**i**) 5 or (**j**) 7 mg/kg furimazine by IG administration at 2 or 3 h before feeding, and blood glucose levels were profiled at different time periods. All statistical comparisons were performed by a two-tailed *t* test. *P < 0.05, **P < 0.01, ***P < 0.001 versus control. Pink area represents normal blood glucose range. Data in (**b–g**) and (**i–j**) represent the mean ± s.d. from 5 mice. Source data are provided in the Source Data file.

implanted T1D mice showed sustainable restoration of blood glucose for up to 13 days (Fig. 4e), consistent with the results of insulin production upon furimazine induction (Fig. 4f). In contrast, T1D mice implanted with the same microencapsulated cells showed a markedly shortened antidiabetic efficacy upon external blue light illumination (Supplementary Fig. 12), probably due to the low tissue permeability of blue light. Notably, IG administration of furimazine has similar effects of controlling blood glucose homeostasis as IP or IV administration (Fig. 4g), which is a critical favorability criterion because it avoids the pain and anxiety caused by the traditional intramuscular injection of insulin.

Keeping blood glucose levels in the target range during the day is important for T1D patients, especially after food intake[46]. Therefore, T1D patients always need intramuscular injections of insulin before meals to avoid a sharp rise in blood glucose levels (Fig. 4h)[40], which is both inconvenient and painful. Here, we attempted to maintain blood glucose homeostasis by oral administration of furimazine instead of traditional intramuscular injection of insulin before and after meals (Fig. 4h). Briefly, T1D mice with implants were orally administered different dosages of furimazine at 2 h or 3 h before feeding, and blood glucose levels were monitored before and after feeding (Fig. 4i and j). We observed narrow fluctuations in glucose levels before and after feeding compared to the trend in control mice kept only in dark conditions (Fig. 4i and j, Supplementary Fig. 13a and b). Intriguingly, the optimal dosage and time for furimazine administration were closely correlated with the feeding intervals. For example, if the feeding interval was 4 h, administration of 5 mg/kg furimazine 2 h before feeding was optimal; if the feeding interval was 6 h, administration of 7 mg/kg furimazine 3 h before feeding was optimal (Fig. 4i and j). These correlations were mainly contributed by the expression profiles induced by different dosages of furimazine, providing extremely flexible options for potential clinical applications in the future. Further studies showed that oral administration of furimazine twice daily at a dosage of 14 mg/kg for a total of 7 days had little toxicity to the animals, as shown by the little difference in body weight and organ sections between the mice treated with furimazine or not (Supplementary Fig. 14). Notably, a single dose of up to 280 mg/kg furimazine by oral administration also showed little toxicity (Supplementary Fig. 15). Collectively, LuminON system can serve as a valuable and safe tool for precise controllable cell therapy.

## Discussion

BRET-based technologies have been widely used for real-time detection of protein-protein interactions[47] and for construction of biosensors[48–50] and bioluminescent-fluorescent proteins[51]. In 2013, BRET technology was first applied in optogenetics to engineer luminopsins, the fusion proteins of luciferase and opsin[15,16]. Unlike traditional optogenetic approaches, the use of luminopsins allows interrogation of neuronal circuits at different temporal and spatial resolutions by choosing either extrinsic physical light or intrinsic biological light for their activation[15,16]. Compared to the luciferases (GLuc or RLuc) used in luminopsins, Nluc has the advantages of high brightness, no requirement of ATP and resistance to autoinhibition by its catalytic byproducts[32,52]. In particular, the emission spectrum of Nluc perfectly matches the absorption spectra of the light-sensitive LOV proteins. All these favorable properties enable Nluc to be an ideal luminescence donor to activate LOV protein-based optogenetic systems via BRET. Very recently, a transcription factor SPARK2 that incorporates Nluc to control the light-responsive *Avena sativa* LOV domain (AsLOV2) was engineered to detect protein-protein interactions[53]. Moreover, a Nluc-based bioluminescent-fluorescent protein (CeNLuc) was

engineered to active a series of blue-green optogenetic systems[51]. Intriguingly, Nluc in both the systems was non-fused to the photosensors, demonstrating that BRET may also occur between Nluc and its non-fused partner photosensor when they are in high concentrations. However, Nluc in these systems does not directly activate the transcription factor, which might lead to delay or low efficiency in transcriptional control. In addition, both systems consisted of multiple protein components, increasing the technical complexities and limiting the portability of these systems. Furthermore, whether these systems allow pulsatile regulation of transgene expression remains to be further studied. In this study, we built a BRET-based transgene expression system in a straightforward manner by directly fusing Nluc to a light-switchable transcription factor. Upon induction by furimazine, the luminescence emitted from Nluc directly activated the proximal VVD domain in luminGAVPO, resulting in dimerization of luminGAVPO and binding of luminGAVPO to its cognate promoter, thereby initiating transcription of the target gene. When furimazine is consumed, the luminGAVPO dimer gradually dissociates from the promoter, leading to inactivation of transcription. Therefore, furimazine-induced transgene expression mediated by luminGAVPO is pulsatile. More importantly, the luminGAVPO-based LuminON system consists of only one integrated genetic cassette, conferring it with all the advantages of single-component optogenetic systems, including speed, simplicity, and versatility[54].

The pulsatile expression profile mediated by the LuminON system would be especially useful for the in vivo study of pulse behavior in naturally occurring systems to understand the principles and kinetics underlying such spatiotemporal patterns in gene expression. For example, insulin secretion is normally maintained at low levels in normal conditions but quickly rises after food intake. In type 1 diabetic patients, blood glucose homeostasis is disrupted since the body does not produce enough insulin[40]. Patients usually administer insulin through various external sources, such as insulin jet injectors and insulin syringes[55]. The proof-of-concept studies described here show the use of the LuminON system to control the pulsatile and repetitive expression of insulin to enhance blood-glucose homeostasis in T1D mice, which could prevent the hypoglycemia caused by excessive insulin from continuous production by traditional chemical-inducible systems. In particular, since both the pulse amplitude and duration of transgene expression are highly tunable via adjustment of the amounts of furimazine, this system is applicable for T1D patients with different dietary habits, e.g., 3 or 4 meals a day. Additionally, due to the design principle of luminGAVPO, we can further fine-tune the pulse amplitude and duration by modulating the photoadduct decay of the VVD domain[20]. In principle, VVD mutants with faster photoadduct decay may enable a narrower amplitude and shorter pulse duration of the system, as these mutants can accelerate recovery of the activated luminGAVPO to the inactivated state, resulting in earlier termination of transcription. Therefore, it is possible to realize more precise control of the pulse amplitude and duration for the study of various pulsatile dynamics of different pulsing systems simply by combining different furimazine amounts with different luminGAVPO mutants.

The Nluc bioluminescence system derives from deep sea shrimp and thus is heterogeneous and orthogonal in mammals, indicating the relatively low possibility of cross-talk between the host chassis and the transcription factor or inducer. Furthermore, previous studies have shown that furimazine can cross the blood-brain barrier[56], indicating the potential for application of the LuminON system in pulsatile regulation of transgene expression in the brain. All these favorable properties would be useful for expanding the applications of the LuminON system in the future.

In our study, the engineered cells were microencapsulated and implanted intraperitoneally, which allowed free diffusion of substances with low molecular weights (<72 kDa) while shielding their cellular content from physical contact with the host's immune system[42]. However, implantation of the microencapsulated cells may also elicit innate immune-mediated foreign body responses that result in fibrotic deposition, nutrient isolation, and donor tissue necrosis[57,58]. In order to improve the safety and durability of LuminON system-based gene therapy, the virus vector, e.g., adeno-associated virus that has been approved for gene therapy by FDA[59], can provide an alternative approach to deliver our LuminON system into animals for long-term gene therapy.

Overall, the BRET-based optogenetic device developed in this study can be used to precisely control the dynamics of key signaling proteins in a pulsatile fashion both in vitro and in vivo and can serve as a model system for improved understanding of the principles and kinetics underlying similar pulsing behaviors found in nature, for engineering precise controllable cellular systems in the synthetic biology field, and for providing methodologies for pharmacological interventions.

## Methods

**Cell culture and transfection.** HEK293, HEK293T, HeLa, COS-7 and U-87 cells (Cell Bank of Chinese Academy of Science) were maintained in Dulbecco's modified Eagle's medium (high glucose) (HyClone) supplemented with 10% FBS (Biological Industries) and 1% penicillin-streptomycin 100× solution (HyClone). PC-3, A549 and H1299 cells were grown in RPMI 1640 (HyClone) supplemented with 10% FBS and 1% penicillin-streptomycin 100× solution. All cells were cultured at 37 °C in a humidified atmosphere of 95% air and 5% CO₂. Cells were plated in antibiotic-free DMEM (high glucose) or RPMI 1640 supplemented with 10% FBS 16-24 h before transfection.

We typically used Hieff Trans™ liposomal transfection reagent (YEASEN, 40802ES02) for transfection according to the manufacturer's protocol. In detail, cells were seeded at 70–90% confluence before transfection. For 96-well plates, 0.1 µg of DNA (0.05 µg of transactivator plasmid and 0.05 µg of reporter plasmid) and 0.25 µl of Hieff Trans™ liposomal transfection reagent were diluted in 25 µl of Opti-MEM (GIBCO) and then mixed. The DNA-lipid complex was added to the cells after a 20-min incubation at room temperature.

**Light irradiation and furimazine induction.** Unless indicated, the transfected cells were kept in the dark for 24 h, and then, the medium was replaced with fresh complete serum/antibiotic-containing medium (no phenol red). Then, the transfected cells were illuminated by 1.4 W·m⁻² (average irradiance) blue light from an LED lamp (460-nm peak) from below or remained in the dark with or without furimazine before characterization. The LED lamps were controlled with a timer to adjust the overall dose of blue light illumination during the specified period. Neutral density filters were used to adjust the light irradiance. Furimazine (synthesized by ChemPartner) was dissolved in 85% ethanol and 15% glycerol at a concentration of 5 mM or dissolved in DMSO at a concentration of 100 mM.

**Plasmid construction.** For mammalian expression of the LuminON system, the coding sequence of Nanoluc was cloned into pGAVPO and pSV40-GAVPO[27] using the Hieff Clone® Plus One Step Cloning Kit (YEASEN) to obtain plasmids expressing different configurations of Nluc-GAVPO fusion proteins driven by the CMV or SV40 promoter. For application of the LOVTRAP or CRY2-CIB1 system, Nanoluc was ligated into the pTriEx-mCherry-LOV2 (Addgene: 81041) plasmid digested by Acc65I and NheI (Thermo Scientific), the pCIBN(deltaNLS)-pmGFP (Addgene: 26867) plasmid digested by Eco47III and AgeI (Thermo Scientific), or the pCRY2FL(deltaNLS)-mCherryN1 (Addgene: 26871) plasmid digested by Eco47III and XhoI (Thermo Scientific). For cell therapy of the LuminON system in mice, luminGAVPO was ligated into the pYH88 plasmid digested by Eco47III and BsrGI, and UAS_G-TATA-Gluc-P2A-mINS was ligated into the pWS251 (obtained from Haifeng Ye, ECNU) plasmid digested by NotI and MluI.

**Chemiluminescence and absorption assays.** A Synergy 2 or a Neo2 multi-mode microplate reader (BioTek) were used to measure the chemiluminescence of the samples. A Gaussia luciferase assay kit was used to assay the activity of secreted Gluc in cell culture supernatants or mouse blood according to the manufacturer's protocol (NEB). Ten-microliter cell culture supernatants were transferred to each well of a white 384-well plate (Greiner), and 10 µl of coelenterazine solution (1 µM coelenterazine, 0.1 M Tris-HCl buffer at pH 7.4 and 0.3 M sodium ascorbate) was added. Light emission was recorded as relative light units. Furimazine from the Nano-Glo® Luciferase Assay Kit (Promega) was used to detect Nluc activity in cells.

**Quantitative PCR with reverse transcription (RT-qPCR).** The RT-qPCR was carried out according to the method described previously[27]. In detail, total RNA was isolated from the transfected HEK293 cells using a total RNA extraction kit (Promega) according to the manufacturer's instructions. In total, 500 ng of the total RNA was converted to single-stranded cDNA using GoScript Reverse Transcription System (Promega). For real-time qPCR, 2 µl of the cDNA was used for the assay with GoTaq qPCR Master Mix (Promega) and specific primers in Supplementary Table 2 according to the manufacturer's recommendations, by using a qTower cycler (Analytik Jena AG). The specificity of amplification was verified by melting-curve analysis, and the data were collected using qPCRsoft software. Amplification conditions were one cycle of 95 °C for 2 min followed by 40 cycles of 95 °C for 15 s and 60 °C for 60 s, with a final melting-curve analysis step (heating the PCR mixture from 65 °C to 95 °C by 0.5 °C every 5 s) to confirm specificity of amplification and lack of primer dimers. All samples were normalized to the β-actin values and the results expressed as fold changes of cycle threshold (Ct) values relative to the samples before furimazine induction by using the $2^{-\Delta\Delta Ct}$ formula.

**Cell viability assay.** HEK293 cells ($1 \times 10^4$) were seeded into each well of a 96-well plate and transfected with CMV-Nluc or empty plasmid. Twenty-four hours after transfection, the cells were exposed to different concentrations of furimazine for 48 h, and cell viability was evaluated using Cell Counting Kit-8 (Dojindo Laboratories, Gaithersburg, MD) according to the manufacturer's protocol. After treatment with CCK8 at 37 °C for 2 h, the absorbance at 450 nm was measured using the Synergy 2 or the Neo2 multi-mode microplate reader (BioTek).

**Western blot.** For the Western blot analyses, cells were lysed in 1× SDS sample buffer supplemented with a protease/phosphatase inhibitor cocktail (Cell Signaling Technology). Equal amounts of total protein (30-60 µg) were separated by sodium dodecyl sulfate-polyacrylamide gel electrophoresis (SDS-PAGE) and electro-transferred onto PVDF membranes. Membranes were incubated with primary antibodies against FLAG (1:1000, Anti-OctA-Probe, Santa Cruz Biotechnology) and β-actin (1:1000, Anti-β-Actin, Santa Cruz Biotechnology), and then with secondary antibodies (1:3000, Anti-rabbit IgG, HRP-linked Antibody, Cell signaling technology) conjugated to horseradish peroxidase, followed by addition of chemiluminescence detection mixture (YEASEN, 36208ES60) and imaging. The original WB figure of luminGAVPO was shown in Supplementary Note 2.

**Imaging.** Unless indicated, confocal imaging for live-cell fluorescence was performed using a Leica TCS SP8 DIVE microscope equipped with an HC PL APO CS2 ×63.0/1.40 OIL objective and an HyD detector. BFP fluorescence was imaged using a 405-nm laser and an emission wavelength range of 415–475 nm. mCherry fluorescence was imaged using a 561-nm laser and an emission wavelength range of 570–650 nm. For imaging of mCherry fluorescence in mouse livers, the livers were dissected from the sacrificed mice 18 h after intravenous injection of plasmids. Images were acquired using an In-Vivo Multispectral System FX (CareStream) with an excitation wavelength of 600/20 nm and an emission wavelength of 670/50 nm. The mCherry fluorescence was resolved from background fluorescence by the CareStream Multispectral program.

Leg skeletal muscles were isolated from the sacrificed mice and sectioned using a Leica VT1200 S fully automated vibrating-blade microtome (Leica, Co., Ltd.). The parameters were 3 mm amplitude, 1 mm/s speed and 300 µm step size (thickness). Muscle sections were maintained in ice-cold PBS-G and then placed evenly between two glass cover slips for imaging. The mCherry fluorescence was detected by a Leica TCS SP8 DIVE microscope equipped with an HC Plan APO CS2 20×/0.75 NA dry objective and an HyD detector, using a 561-nm excitation laser and an emission wavelength range of 570–650 nm.

**Generation of stable cell lines.** The polyclonal HEK_FUR-Gluc-P2A-mINS population, transgenic for furimazine-inducible Gluc and insulin expression, was constructed by cotransfecting $5 \times 10^4$ HEK293T cells with 100 ng of pWS251 (ITR-pU5-Gluc-P2A-mINSpA:PmPGK-ZeoR-P2A-EGFP-pA-ITR), 100 ng of pYH88 (ITR-P_CMV-NG:PmPGK-PurR-pA-ITR) and 20 ng of the Sleeping Beauty transposase expression vector pCMV-T7-SB100 (P_CMVSB100X-pA)[41]. After selection with 1 µg/mL puromycin and 100 µg/mL zeocin for 2 weeks, the surviving polyclonal population of HEK_FUR-Gluc-P2A-mINS was seeded into monoclonal cells in a 12-well plate for further cultivation. After 2 weeks, the monoclonal cells reached 40–50% confluency and were kept in the dark with or without 2.5 µM furimazine for 24 h, and then, the Gluc activity was quantified. HEK_FUR-Gluc-P2A-mINS cells with the highest sensitivity to furimazine and low background expression in the dark were used for the following studies. Stable cell lines were regularly tested for the absence of mycoplasma and bacterial contamination.

**Mice experiments.** All procedures involving animals were approved by the Institutional Animal Care and Use Committee of Shanghai and were conducted in accordance with the National Research Council Guide for Care and Use of Laboratory Animals. KM mice (Shanghai Jie Si Jie Laboratory Animal CO. LTD) or C57BL/6 J mice (SLRC Laboratory Animals) were used for animal experiments. Mice were reared in ECNU (East China Normal University) Laboratory Animal Center.

To induce gene expression in mice livers, Kunming (KM) mice were subjected to intravenous coinjection of 100 µg of pU5-mCherry or pU5-Gluc and 10 µg of pSV40-NG in 2–3 ml (12% of the body weight in grams) of Ringer's solution (147 mM NaCl, 4 mM KCl, 1.13 mM CaCl$_2$) within 5–7 s, and then, the mice were kept in the dark. Six hours after plasmid injection, the mice were administered 5 mg/kg furimazine dissolved in phosphate-buffered saline (PBS) via intravenous injection or were administered 7 mg/kg furimazine dissolved in PBS orally. For the blood Gluc reporter assay, blood samples were collected from these mice by making a small incision in the tail and directly adding it to an Eppendorf tube containing EDTA as an anticoagulant (10 mM final concentration). The samples were immediately centrifuged at $1500 \times g$ for 5 min and the supernatants (blood serum) were transferred and stored at $-80$ °C before measurement. The Gluc activities in the supernatants were measured using a Gaussia luciferase assay kit as described above.

To induce gene expression in mouse muscles, KM mice were electroporated with 50 µg of plasmids with the LuminON system. Electrode needles were inserted into the muscle to a depth of 5 mm together with the syringe. A total of 50 µg of plasmids for the LuminON system (4.5 µg of CMV-NG, 45.5 µg of pU5-mCherry) at 1 µg/µL in ddH$_2$O were then injected into the leg skeleton muscles of mice with a syringe. Electric pulses were delivered immediately using an in vivo electroporator (TERESA-EPT I, Teresa) after removing the syringe with the following parameters: voltage, 60 V; pulse width, 50 ms; frequency, 1 Hz. Six hours after electroporation, the mice were induced by furimazine via tail intravenous injection; control mice were kept in the dark. Twenty-four hours after induction, leg skeletal muscle was isolated immediately after the mice were sacrificed and then sectioned with a Leica VT1200 S fully automated vibrating-blade microtome for imaging.

Construction of the type 1 diabetic mouse model (T1D) was carried out using previously described procedures[60]. Briefly, ten-week-old male mice (C57BL/6 J, East China Normal University Laboratory Animal Centre) were fasted for 16 hs and received intraperitoneal injection of streptozotocin (STZ, 50 mg/kg in 0.1 M citrate buffer, pH 4) daily for 5 days. Two weeks later, fasted mice with blood glucose levels over 15 mmol/L were chosen as diabetic mice and used for further experiments.

For microcapsule implants, intraperitoneal implants were produced by encapsulating transgenic HEK293T cells into coherent alginate-poly-(L-lysine)-alginate beads (400 µm; 200 cells/capsule) using a B-395 Pro encapsulator (BÜCHI Labortechnik) set to the following specifications: a 200-µm nozzle with a vibration frequency of 1,300 Hz, a 25-mL syringe operated at a flow rate of 450 U, and 1.10-kV voltage for bead dispersion[60–62]. C57BL/6 J T1D mice were intraperitoneally injected with 800 µL of PBS containing $4 \times 10^6$ microencapsulated transgenic cells (200 transgenic HEK293T cells/capsule). For the glucose tolerance test, 48 h after implantation, fasted mice (16 h) received an intraperitoneal injection of aqueous 1 g/kg D-glucose, and the glycemic profile of each animal was tracked via tail vein blood samples within 120 min. Blood glucose levels of the mice were measured using a commercial glucometer (Contour Next; Bayer HealthCare; detection range: 0.5–35 mM). Insulin in mouse serum was quantified using a mouse insulin ELISA Kit (Mercodia AB; cat. no. 10-1247-01) according to the manufacturer's instructions.

To study the toxicity of furimazine in vivo, animals were administrated with different dosages of furimazine twice daily in the range of 0–14 mg/kg, via intravenous (IV), intraperitoneal (IP) or intragastric (IG) administration, respectively. Animal body weight was monitored every day for a total of 7 days. The mice were then sacrificed by cervical dislocation and internal organs (heart, liver, lung, and kidney) were removed and fixed in 4% formalin equilibrated with PBS (pH 7.4). Pieces of fixed internal organs were dried in alcohols of increasing concentration and embedded in paraffin. 3 µm-thick histology sections were obtained using a Leica RM2016 and imaged using a Nikon Eclipse E100 microscope equipped with a Nikon digital sight DS-FI2. To test the maximum tolerated dosage of furimazine by oral administration, a dosage of up to 280 mg/kg furimazine was intragastrically administered. Twenty-four hours after administration, the sections of internal organs (heart, liver, lung and kidney) were obtained and analyzed as described above.

**Statistical analysis**. For experiments with duplicates, the results were shown as means ± s.d., unless stated otherwise. For comparison of furimazine induced mCherry expression in the mouse tibialis posterior muscles, furimazine induced pulsatile expression of insulin to enhance blood glucose homeostasis in T1D mice, analysis was performed by a two-tailed Student's $t$ test (Figs. 1c and 1e–f, Fig. 3e–g, Figs. 4b–g, 4i–j, Supplementary Fig. 1e, Supplementary Fig. 3a–c, Supplementary Fig. 4a–b, Supplementary Fig. 7, Supplementary Fig. 9, Supplementary Fig. 11b, Supplementary Fig. 12, Supplementary Fig. 13, Supplementary Fig. 14a, d and g). All the $P$ values in the figures were listed in Supplementary Table 1.

**Reporting summary**. Further information on research design is available in the Nature Research Reporting Summary linked to this article.

## Data availability

All relevant data supporting the findings of this study are available within the paper and its supplementary information files. The data that support the findings in this study are available upon reasonable request from the corresponding authors. Source data are provided with this paper.

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

## Acknowledgements

We thank Dr. Huan Xu at East China University of Science and Technology, Jianli Yin, Shuai Xue, Guiling Yu, Yang Zhou at East China Normal University for technical assistance. This research was supported by the National Key Research and Development Program of China (2019YFA0904800 and 2017YFA050400 to Y.Y., 2019YFA0904500 to H.Y.), NSFC (91857202 and 21937004 to Y.Y., 31971349 and 31600688 to X.C. and 31901033 to T.L.), the Shanghai Science and Technology Commission (18JC1411900 to Y.Y., 19ZR1472800 to X.C. and 18JC1411000 to H.Y.), the Shanghai Rising-Star Program to X.C., the Young Elite Scientists Sponsorship Program by CAST to X.C, the Shanghai Sailing Program (19YF1411300 to T.L.), the Fundamental Research Funds for the Central Universities to Y.Y. and X.C., the China Postdoctoral Science Foundation (2019M651413 to T.L.).

## Author contributions

Concepts were conceived by Y.Y., H.Y. & X.C; Y.Y., H.Y., X.C. & T.L. designed the experiments and analyzed the data; T.L. performed live cells experiments. T.L., X.L and Y.Q. performed plasmid construction. T.L., J.S. and S.L. performed mice experimental. L.Z. and Y.Z. gave technical support and conceptual advice. Y.Y., H.Y., X.C. and T.L. wrote the manuscript.

## Competing interests

Y.Y., X.C., and T.L. are named inventors of patent application no. 2020115018802. The remaining authors declare no competing interests.
