## [Peer Review File · Nature Communications]

Reviewers' Comments:

Reviewer #1:

Remarks to the Author:

In this manuscript, the authors reported a synthetic BRET-based transgene expression (LuminON) system to achieve pulsatile and quantitative activation of transgene expression via both chemogenetic and optogenetic approaches in vitro and in vivo. The system is based on a luminescent transcription factor, termed luminGAVPO, by fusing NanoLuc luciferase to the light-switchable transcription factor GAVPO. The amplitude and duration of transgene expression can be tunable via adjustment of the amount of furimazine or external light. In type 1 diabetic mice, the implanted microcapsules with engineered HEK cells can achieve LuminON-mediated blood-glucose homeostasis. Overall, the approach is highly innovative and the systems are well designed and characterized. I recommend this article be accepted for publication after the authors address the following issues:

1. It is found that Gluc expression slightly decreased when the furimazine concentration was higher than 5 μ M. The authors are suggested to provide a potential explanation for this phenomenon.
2. "Highly precise control of transgene expression was achieved by combining the furimazine concentration and light irradiance." While the combination of furimazine and light could achieve better control of gene expression, this approach may be limited to in vitro or ex vivo set-ups. The authors should comment more on it.
3. Cell-loaded microcapsules were implanted intraperitoneally. The authors are suggested to discuss the potential downsides of intraperitoneal implantation and provide some alternative approaches.
4. Toxicity concerns associated with furimazine in vivo should be tested. Maximum tolerance dosage of furimazine through different administration approaches can be provided.
5. A line to indicate the threshold of normal/hyperglycemia is suggested to add in Figure 4.
6. Statistical analyses for Figure 1 are suggested to evaluate.
7. Scale bars for Figure 3 should be provided.

Reviewer #2:

Remarks to the Author:

The investigators previously generated a light-sensing transcription factor (GAVPO), based on the VIVID Lov domain. In the current studies they fused a light-emitting luciferase, NanoLuc, to GAVPO (luminGAVPO), allowing activation of the transcription factor by bioluminescence (through BRET). They are applying their transgene expression system (LuminON) specifically to the pulsatile regulation of blood glucose in diabetic mice.

While there are other NanoLuc based transcriptional systems (which the authors reference in their paper), the luminGAVPO tool is novel and has features that will be of interest in many applications across disciplines. The main advantages of the luminGAVPO tool are its very high light sensitivity and its simplicity as a one-component system.

The authors present convincing and high-quality data for optimized configurations of NLuc and GAVPO, the relationship between expression levels and background, titration of furimazine concentrations to achieve optimal background-signal ratio, and toxicity testing of furimazine concentrations (however: only in vitro; see below). The tunability of transcription amplitude and duration, reversability, and possibility of repeated driving of transcription, demonstrated both in cells and in animals, is impressive. Another strength of the tool is the demonstration of intragastric application of luciferase substrate (furimazine), in addition to intravenous and intraperitoneal routes. This is especially attractive with future clinical applications in mind.

There are some issues that should be addressed:

1. The authors use a clever design by employing secreted Gaussia luciferase (GLuc) as the reporter for furimazine-NLuc bioluminescence-driven transcription. While furimazine cannot be used as a substrate by GLuc, coelenterazine (the substrate for GLuc) can be used by NLuc, though with low efficiency. Furthermore, only GLuc is secreted, while NLuc-GAVPO is not expected to be in the cell supernatant. Nevertheless, it would be good to see, at least in one in vitro experiment, supernatants measured for GLuc activity also measured for light emission (or absence thereof) with furimazine.

2. The authors conclude that luminGAVPO needs to be a fusion protein that works through bioluminescence resonance energy transfer (BRET) by demonstrating that co-expression of the two moieties does not work. However, this is concentration dependent: higher concentrations of the two partners will increase the chance of BRET. This is likely how interactions between NanoLuc and their respective non-fused photosensors in SPARK2 and LumiFluors work. This should be included in the text/discussion.

3. The authors compare the halftimes of pulsatile transcription using luminGAVPO to chemically induced systems that have much longer halftimes. The data are clear – however, the authors might want to point out that this in vitro comparison does not necessarily translate to the in vivo situation, where the halftime of the various compounds can differ vastly.

4. The authors show convincing applications of luminGAVPO in diabetic mice. However, the furimazine concentrations are fairly high (5 – 7 mg/kg). Compared to doses used in imaging studies, furimazine concentrations here are 20 times higher (0.25 mg/kg: Stacer, A.C., Nyati, S., Moudgil, P., Iyengar, R., Luker, K.E., Rehemtulla, A., and Luker, G.D., *Mol. Imaging*, 2013, vol. 12, pp. 1–13). Even compared to a toxicity study with furimazine they are still high (1 mg/kg: Shipunova, V.O., Shilova, O.N., Shramova, E.I. et al. A Highly Specific Substrate for NanoLUC Luciferase Furimazine Is Toxic in vitro and in vivo. *Russ J Bioorg Chem* 44, 225–228, 2018). While this is not a detriment, the authors should put the concentrations they used in context, especially as they did not address toxicity in vivo (only in cell assays).

5. Figure 4 needs some more explanation/discussion regarding the 2-week application of furimazine. First, there is no effect on blood glucose levels on day 1 – is this due to necessary build-up of insulin levels from the transgene? Second, the effect is gone by day 14 – is this due to the implanted HEK cells disappearing or is there resistance of the luminON system developing over time?

6. The authors refer to control mice kept only in dark conditions: does this mean 'no furimazine' and/or mice were physically kept in the dark to avoid ambient light exposure (which might be necessary with highly light-sensitive photoreceptors)?

7. Legend of Supplementary Figure 1: "(d) Imaging and (e) quantification of furimazine-induced translocation of NLuc-mCherry-LOV2 to the outer membrane of mitochondria" – should say from the outer membrane of mitochondria to the cytoplasm and return to the outer membrane of mitochondria as light dims to the dark state.

8. Check references:

87 opsins (ref 15, 16). Thus, these luminescent opsins allowed regulation of neuron activity in a pulsatile fashion (ref 19, 20 ?). We therefore hypothesized that similar concepts could be used

292 luminopsins, the fusion proteins of luciferase and opsin (ref 43, 44 ?).

Compared to the luciferases (GLuc or RLuc) used in

296 luminopsins, Nluc has the advantages of high brightness, no requirement of ATP and

Luminopsin using RLuc is described in Tung 2015 – this reference is missing:

Tung JK, Gutekunst CA, Gross RE. Inhibitory luminopsins: genetically-encoded bioluminescent opsins for versatile, scalable, and hardware-independent optogenetic inhibition. *Sci Rep.* 2015;5:14366.

9. Source of furimazine should be disclosed:

623 used to adjust the light irradiance. Furimazine was dissolved in 85% ethanol and 15% glycerol 624 at a concentration of 5 mM.

With the few additions regarding methodology added the studies could be reproduced by other researchers.

In summary, based on the overall quality of the presented data and the unique features of the novel tool this reviewer feels that the paper will have a considerable influence in the field of bioluminescent control tools and their applications in several biosciences.

Reviewer #3:

Remarks to the Author:

Review comments:

In this manuscript Ting Li et al describe the development of a mechanism combining pulsing cellular mechanism for the control of gene expression allowing pulsatile regulation of the transgene in vitro and in vivo in term of leading to a pulsatile and quantitative activation of transgene expression in both mammalian cells and animals in a furimazine- or light irradiance-dependent manner; a technology developed based on a luminescent transcription factor termed luminGAVPO . They have also tested luminGAVPO in vivo using an experimental murine model of hyperglycemia induced with low dose injections of streptozotocin (MLD-STZ) and they demonstrated that the implantation of stably engineered cells of luminGAVPO-mediated insulin production provided a better control of blood glucose MLD-STZ mice. This is an interesting manuscript, although not being the first study to show a proof of concept highlighting the usefulness of synthetic light-pulsing transcription mechanism as similar mechanism was used by Haifeng Ye et al applying synthetic light-pulsing transcription to GLP1 allowing it to be remote-controlled in vitro as in vivo and attenuate hyperglycemia in murine model of T2D (Haifeng Ye et al, *Science* 2011). So the authors here are presenting another analogue technology which although seems attractive as it steps away from predominantly pill-based approaches towards much more targeted therapy by providing control over timing, location and delivered dosage of biologic drugs and all the conclusions drawn by the authors are clearly supported by data, but few comments should be addressed:

Minor comments:

1- It has been reported that engineering of stably transfected mammalian cells for later implantation although seems relevant to future clinical applications, but also faces several a challenges because viral vectors may require specific surface receptors for entry or can be immunogenic or cause insertional mutation, please comment on this.

3- One minor concern about the animal model used by the authors and named as T1D mouse model, which is more an experimental model of hyperglycemia, although multiple low dose)-STZ injections for several consecutive days are often used to model T cell dependent autoimmune destruction of pancreatic β cells but doesn't fully mimic the T1D model. On the other hand, is there any concern regarding the use of the common and known T1D model: the NOD mouse? Please comment.

4- Using this STZ-low dose model, researchers have found that GLUT-2 was a main target for STZ-

toxicity as significant reduction of GLUT-2 protein preceding the onset of hyperglycemia in mice, did the authors thought about measuring GLUT-2 following the infusion of lumenGAVPO in mice? Please comment.

5- did the authors thought about assessing insulin response to nonglucose secretagogue such arginine?

Response to the reviewers

We would like to thank all of the referees for their highly constructive comments. As we hope you will agree, the careful revision process we have undertaken has substantially improved both the scientific rigor and impact of our study. We present point-by-point responses to each of the referee comments (below). We therefore invite you to examine our responses below, and we would again like to thank all of the referees in the editor for their ongoing work on our behalf.

Reviewers' Comments to Author

(Line numbers mentioned in a report may not coincide with the original line numbers.)

Reviewer #1 (Remarks to the Author):

In this manuscript, the authors reported a synthetic BRET-based transgene expression (LuminON) system to achieve pulsatile and quantitative activation of transgene expression via both chemogenetic and optogenetic approaches in vitro and in vivo. The system is based on a luminescent transcription factor, termed luminGAVPO, by fusing NanoLuc luciferase to the light-switchable transcription factor GAVPO. The amplitude and duration of transgene expression can be tunable via adjustment of the amount of furimazine or external light. In type 1 diabetic mice, the implanted microcapsules with engineered HEK cells can achieve LuminON-mediated blood-glucose homeostasis. Overall, the approach is highly innovative and the systems are well designed and characterized. I recommend this article be accepted for publication after the authors address the following issues:

Thank you very much for your highly positive comments as well as the highly constructive suggestions to help increase the quality of our manuscript.

Comments for the Authors

1. It is found that Gluc expression slightly decreased when the furimazine concentration was higher than 5 μ M. The authors are suggested to provide a potential explanation for this phenomenon.

Response: The reviewer raised an interesting phenomenon that Gluc expression from LuminON system slightly decreased when the furimazine concentration was higher than 5 μ M. In the revised manuscript, we tested Gluc expression from a constitutive CMV promoter in the cells cultured with different concentrations of furimazine. Our results showed that Gluc activity also slightly decreased when the furimazine concentration was higher than 5 μ M (**Supplementary Fig. 9a**). It seemed that high concentration of furimazine could slightly inhibit protein synthesis, but little effect on cell viability was observed (**Supplementary Fig. 9b**).

We have integrated new results into the revised manuscript and greatly appreciate the reviewer's valuable suggestion.

2. "Highly precise control of transgene expression was achieved by combining the furimazine concentration and light irradiance." While the combination of furimazine and light could achieve

better control of gene expression, this approach may be limited to in vitro or ex vivo set-ups. The authors should comment more on it.

Response: The reviewer raised the concern whether the combination of furimazine and light irradiance could achieve highly precise control of gene expression in vivo. Our previous data have shown that different furimazine concentrations could result in graded activation of transgene expression in mice (**Fig. 3f**). In addition, light irradiance can be adjusted by controlling the light intensity or illumination duration. Therefore, precise control of transgene expression in vivo can also be achieved by combining the furimazine concentration and light irradiance. Furthermore, the dual inputs (furimazine and light) also support the development of conceptually complex applications, including for example the OR logic gates.

We have integrated above description into the revised manuscript and appreciate the reviewer's valuable suggestion.

3. Cell-loaded microcapsules were implanted intraperitoneally. The authors are suggested to discuss the potential downsides of intraperitoneal implantation and provide some alternative approaches.

Response: We thank the reviewer for the constructive suggestion. We have briefly discussed the potential downsides of intraperitoneal implantation and provide some alternative approaches in the revised discussion part.

In our study, the engineered cells were microencapsulated and implanted intraperitoneally, which allowed free diffusion of substances with low molecular weights (<72 kDa) while shielding their cellular content from physical contact with the host's immune system¹. Alginate-poly-L-lysine-alginate-based encapsulation, as used in our study, has been successfully validated in human clinical trials (ClinicalTrials.gov NCT01379729)² and the performance of the material is continuously improved for clinical applications^{3,4}. Soon-Shiong P et al. reported the successful transplant of human islets encapsulated in alginate-poly-L-lysine-alginate microcapsules into type 1 diabetic patient⁵.

However, implantation of the microencapsulated cells may elicit innate immune-mediated foreign body responses (FBR) that result in fibrotic deposition, nutrient isolation, and donor tissue necrosis^{6,7}. In addition, the semipermeable membrane of the microcapsule can delay but not completely prevent host immune rejection of xenogeneic grafts⁸; thus, most of the microencapsulated cells would be attacked and damaged by the animal's immune system and lost the function from several weeks to months after implantation. In order to improve the safety and durability of LuminON system-based gene therapy, the virus vector, e.g., adeno-associated virus that has been approved for gene delivery the treatment of human diseases by FDA⁹, can provide an alternative approach to deliver LuminON system-regulated transgene into animals for long-term gene therapy.

4. Toxicity concerns associated with furimazine in vivo should be tested. Maximum tolerance dosage of furimazine through different administration approaches can be provided.

Response: We thank the reviewer for the valuable suggestion. In the revised manuscript, we have tested the toxicity of furimazine in mice. Animals were administrated with different dosages of furimazine twice daily in the range of 0-14 mg/kg, via intravenous (IV), intraperitoneal (IP) or intragastric (IG) administration, respectively. Animal body weight was monitored every day for a total of 7 days. These results showed that no statistical difference in body weight change was observed among the animals treated with 0, 7 and 14 mg/kg furimazine by IP and IG administrations

(**Supplementary Fig. 16a and c**). However, for IV injection, mice treated with 7 mg/kg furimazine showed marked decrease in body weight change compared to the control mice (**Supplementary Fig. 16e**), while all mice treated with 14 mg/kg dose died within two days. Pathological review of sections of heart, liver, lung and kidney also showed no signs of cytotoxicity for IP and IG administrations, but marked toxicity to the kidney for IV injection was observed (**Supplementary Fig. 16 b, d and f**). These results demonstrate that IP and IG administration enable the mice to tolerate much higher dosage of furimazine than IV. Therefore, in this study, we maintained blood glucose homeostasis by oral administration of furimazine instead of traditional intramuscular injection of insulin, which should be much safer, more convenient and less painful. We then tested the maximum tolerated dosage of furimazine by oral administration. The results showed that 140 mg/kg and 280 mg/kg furimazine showed slight toxicity to the livers and/or the kidney of the mice (**Supplementary Fig. 17**). However, up to 280 mg/kg furimazine was not lethal to the mice, no mice died within 24 h after oral administration. Notably, the dosage of 70 mg/kg furimazine, 10 times that used in our animal experiments (7 mg/kg), showed no detectable toxicity to the mice (**Supplementary Fig. 17**). Collectively, LuminON system can serve as a valuable and safe tool for precise controllable cell therapy.

We have integrated above results into the revised manuscript, and greatly appreciate the reviewer's valuable suggestion.

5. A line to indicate the threshold of normal/hyperglycemia is suggested to add in Figure 4.

Response: We have added a line to indicate the threshold of normal and greatly appreciate the reviewer's valuable suggestion.

6. Statistical analyses for Figure 1 are suggested to evaluate.

Response: We have added the statistical analysis in the revised Figure 1 and listed the *P* values in Supplementary Table 1 in the revised manuscript.

7. Scale bars for Figure 3 should be provided.

Response: We have provided scale bars in the revised Figure 3.

Reviewer #2 (Remarks to the Author):

The investigators previously generated a light-sensing transcription factor (GAVPO), based on the VIVID Lov domain. In the current studies they fused a light-emitting luciferase, NanoLuc, to GAVPO (luminGAVPO), allowing activation of the transcription factor by bioluminescence (through BRET). They are applying their transgene expression system (LuminON) specifically to the pulsatile regulation of blood glucose in diabetic mice.

While there are other NanoLuc based transcriptional systems (which the authors reference in their paper), the luminGAVPO tool is novel and has features that will be of interest in many applications across disciplines. The main advantages of the luminGAVPO tool are its very high light sensitivity and its simplicity as a one-component system.

The authors present convincing and high-quality data for optimized configurations of NLuc and GAVPO, the relationship between expression levels and background, titration of furimazine concentrations to achieve optimal background-signal ratio, and toxicity testing of furimazine concentrations (however: only in vitro; see below). The tunability of transcription amplitude and duration, reversability, and possibility of repeated driving of transcription, demonstrated both in cells and in animals, is impressive. Another strength of the tool is the demonstration of intragastric application of luciferase substrate (furimazine), in addition to intravenous and intraperitoneal routes. This is especially attractive with future clinical applications in mind.

We appreciate your positive comments as well as the highly constructive suggestions.

There are some issues that should be addressed:

1. The authors use a clever design by employing secreted Gaussia luciferase (GLuc) as the reporter for furimazine-NLuc bioluminescence-driven transcription. While furimazine cannot be used as a substrate by GLuc, coelenterazine (the substrate for GLuc) can be used by NLuc, though with low efficiency. Furthermore, only GLuc is secreted, while NLuc-GAVPO is not expected to be in the cell supernatant. Nevertheless, it would be good to see, at least in one in vitro experiment, supernatants measured for GLuc activity also measured for light emission (or absence thereof) light emission.

Response: In the revised manuscript, we measured Gluc and NLuc activity in the supernatant of cells transfection with NLuc-GAVPO and pU5-Gluc or NLuc-GAVPO alone. The results showed that high Gluc activities were only observed in the supernatants of cells transfected with NLuc-GAVPO and pU5-Gluc and cultured upon induction by furimazine and/or light illumination, while dim light emission from NLuc-GAVPO were detected for all the supernatants (**Supplementary Fig. 4**). These data indicate that there is negligible NLuc-GAVPO in the cell supernatants and the interference of NLuc-GAVPO is minimal and ignorable.

We have integrated above results into the manuscript and greatly appreciate the reviewer's valuable suggestion.

2. The authors conclude that luminGAVPO needs to be a fusion protein that works through bioluminescence resonance energy transfer (BRET) by demonstrating that co-expression of the two moieties does not work. However, this is concentration dependent: higher concentrations of the two

partners will increase the chance of BRET. This is likely how interactions between NanoLuc and their respective non-fused photosensors in SPARK2 and LumiFluors work. This should be included in the text/discussion.

Response: We agree with the reviewer's opinion that BRET can happen between the non-fused photosensor and luciferase when they are in high concentration. We have provided the description in discussion part of the revised manuscript and greatly appreciate the reviewer's valuable suggestion.

3. The authors compare the halftimes of pulsatile transcription using luminGAVPO to chemically induced systems that have much longer halftimes. The data are clear-however, the authors might want to point out that this in vitro comparison does not necessarily translate to the in vivo situation, where the halftime of the various compounds can differ vastly.

Response: We agree with the reviewer's opinion that the in vitro comparison does not necessarily translate to the in vivo situation. We have discussed this point in the revised manuscript following the reviewer's guidance.

4. The authors show convincing applications of luminGAVPO in diabetic mice. However, the furimazine concentrations are fairly high (5 – 7 mg/kg). Compared to doses used in imaging studies, furimazine concentrations here are 20 times higher (0.25 mg/kg: Stacer, A.C., Nyati, S., Moudgil, P., lyengar, R., Luker, K.E., Rehemtulla, A., and Luker, G.D., *Mol. Imaging*, 2013, vol. 12, pp. 1–13). Even compared to a toxicity study with furimazine they are still high (1 mg/kg: Shipunova, V.O., Shilova, O.N., Shramova, E.I. et al. A Highly Specific Substrate for NanoLUC Luciferase Furimazine Is Toxic in vitro and in vivo. *Russ J Bioorg Chem* 44, 225–228, 2018). While this is not a detriment, the authors should put the concentrations they used in context, especially as they did not address toxicity in vivo (only in cell assays).

Response: The reviewer raised the concern that furimazine used in diabetic mice was significantly higher than the dose used in imaging studies. In the revised manuscript, we have tested the toxicity of furimazine in mice. Animals were administered with different dosages of furimazine twice daily in the range of 0-14 mg/kg, via intravenous (IV), intraperitoneal (IP) or intragastric (IG) administration, respectively. Animal body weight was monitored every day for a total of 7 days. These results showed that no statistical difference in body weight change was observed among the animals treated with 0, 7 and 14 mg/kg furimazine by IP and IG administrations (**Supplementary Fig. 16a and c**). However, for IV injection, mice treated with 7 mg/kg furimazine showed marked decrease in body weight change compared to the control mice (**Supplementary Fig. 16e**), while all mice treated with 14 mg/kg dose died within two days. Pathological review of sections of heart, liver, lung and kidney also showed no signs of cytotoxicity for IP and IG administrations, but marked toxicity to the kidney for IV injection was observed (**Supplementary Fig. 16 b, d and f**). These results demonstrate that IP and IG administration enable the mice to tolerate much higher dosage of furimazine than IV. Therefore, in this study, we maintained blood glucose homeostasis by oral administration of furimazine instead of traditional intramuscular injection of insulin, which should be much safer, more convenient and less painful. We then tested the maximum tolerated dosage of furimazine by oral administration. The results showed that 140 mg/kg and 280 mg/kg furimazine showed slight toxicity to the livers and/or the kidney of the mice (**Supplementary Fig. 17**). However, up to 280 mg/kg furimazine was not lethal to the mice, no mice died within 24 h after oral administration. Notably, the dosage of 70 mg/kg furimazine, 10 times that used in our animal experiments (7 mg/kg), showed no detectable toxicity to

the mice (**Supplementary Fig. 17**). Collectively, LuminON system can serve as a valuable and safe tool for precise controllable cell therapy.

We have integrated above results into the revised manuscript, and greatly appreciate the reviewer's valuable suggestion.

5. Figure 4 needs some more explanation/discussion regarding the 2-week application of furimazine. First, there is no effect on blood glucose levels on day 1 – is this due to necessary build-up of insulin levels from the transgene? Second, the effect is gone by day 14 – is this due to the implanted HEK cells disappearing or is there resistance of the luminON system developing over time?

Response: T1D mice implanted with the microencapsulated cells were treated with furimazine from day 2, not day 1. Therefore, low insulin levels and high blood glucose levels were observed in day 1. Theoretically, the microcapsule supplies a safe refuge for the implanted xenogeneic cells to escape from recognition and attack by the animal's immune system^{2-4,10}. However, previous studies also showed that the semipermeable membrane of the microcapsule can delay but not completely prevent host immune rejection of xenogeneic grafts⁸; thus, it was possible that most of the microencapsulated cells were attacked and damaged by the animal's immune system and lost the function, which in turn affected their effects for restoration of blood glucose levels 14 days after implantation.

We have indicated the experimental details and integrated above discussion into the revised manuscript and apologize for the confusion.

6. The authors refer to control mice kept only in dark conditions: does this mean 'no furimazine' and/or mice were physically kept in the dark to avoid ambient light exposure (which might be necessary with highly light-sensitive photoreceptors)?

Response: Our previous studies have showed that LightOn system is highly sensitive to light and even can be activated by ambient light^{11,12}. Therefore, to minimize the leakage in non-induced state, the mice were physically kept in the dark to avoid ambient light exposure.

7. Legend of Supplementary Figure 1: "(d) Imaging and (e) quantification of furimazine-induced translocation of Nluc-mCherry-LOV2 to the outer membrane of mitochondria" – should say from the outer membrane of mitochondria to the cytoplasm and return to the outer membrane of mitochondria as light dims to the dark state.

Response: We have revised the figure legend following the reviewer's guidance.

8. Check references:

87 opsins (ref 15, 16). Thus, these luminescent opsins allowed regulation of neuron activity in a 88 pulsatile fashion (ref 19, 20 ?). We therefore hypothesized that similar concepts could be used 292 luminopsins, the fusion proteins of luciferase and opsin (ref 43, 44 ?).

Compared to the luciferases (GLuc or RLuc) used in 296 luminopsins, Nluc has the advantages of high brightness, no requirement of ATP and Luminopsin using RLuc is described in Tung 2015 – this reference is missing: Tung JK, Gutekunst CA, Gross RE. Inhibitory luminopsins: genetically-encoded bioluminescent opsins for versatile, scalable, and hardware-independent optogenetic inhibition. *Sci Rep.* 2015;5:14366.

Response: We have checked all the references throughout the manuscript and greatly appreciate the reviewer's valuable suggestion.

9. Source of furimazine should be disclosed:

623 used to adjust the light irradiance. Furimazine was dissolved in 85% ethanol and 15% glycerol 624 at a concentration of 5 mM.

With the few additions regarding methodology added the studies could be reproduced by other researchers.

Response: We have provided the source of furimazine in the revised manuscript.

In summary, based on the overall quality of the presented data and the unique features of the novel tool this reviewer feels that the paper will have a considerable influence in the field of bioluminescent control tools and their applications in several biosciences.

Response: We thank the reviewer for this enthusiastic comment.

Reviewer #3 (Remarks to the Author):

Review comments:

In this manuscript Ting Li et al describe the development of a mechanism combining pulsing cellular mechanism for the control of gene expression allowing pulsatile regulation of the transgene in vitro and in vivo in term of leading to a pulsatile and quantitative activation of transgene expression in both mammalian cells and animals in a furimazine- or light irradiance-dependent manner; a technology developed based on a luminescent transcription factor termed luminGAVPO . They have also tested luminGAVPO in vivo using an experimental murine model of hyperglycemia induced with low dose injections of streptozotocin (MLD-STZ) and they demonstrated that the implantation of stably engineered cells of luminGAVPO-mediated insulin production provided a better control of blood glucose MLD-STZ mice. This is an interesting manuscript, although not being the first study to show a proof of concept highlighting the usefulness of synthetic light-pulsing transcription mechanism as similar mechanism was used by Haifeng Ye et al applying synthetic light-pulsing transcription to GLP1 allowing it to be remote-controlled in vitro as in vivo and attenuate hyperglycemia in murine model of T2D (Haifeng Ye et al, Science 2011). So the authors here are presenting another analogue technology which although seems attractive as it steps away from predominantly pill-based approaches towards much more targeted therapy by providing control over timing, location and delivered dosage of biologic drugs and all the conclusions drawn by the authors are clearly supported by data, but few comments should be addressed:

We appreciate this reviewer's positive and enthusiastic comments and helpful constructive suggestions.

Minor comments:

1- It has been reported that engineering of stably transfected mammalian cells for later implantation although seems relevant to future clinical applications, but also faces several a challenges because viral vectors may require specific surface receptors for entry or can be immunogenic or cause insertional mutation, please comment on this.

Response: We agree with the reviewer's opinion that viral vectors may require specific surface receptors for entry or can be immunogenic or cause insertional mutation. However, the stably transfected mammalian cells in our study were constructed using the Sleeping Beauty transposon system¹³, not the viral vectors, which may avoid above potential drawbacks for viral vectors.

2- One minor concern about the animal model used by the authors and named as T1D mouse model, which is more an experimental model of hyperglycemia, although multiple low dose)-STZ injections for several consecutive days are often used to model T cell dependent autoimmune destruction of pancreatic β cells but doesn't fully mimic the T1D model. On the other hand, is there any concern regarding the use of the common and known T1D model: the NOD mouse? Please comment.

Response: We agree with the reviewer's opinion that the nonobese diabetic (NOD) mouse is the most widely used for research in Type 1 Diabetes (T1D) as the NOD shares a number of genetic and immunologic traits with the human form of the disease¹⁴. The NOD mouse model has come in multiple forms including identifying key genetic and environmental risk factors and how they may contribute to disease susceptibility and pathogenesis, and also provides insights into the roles of the innate immune cells as well as the B cells in contributing to the T cell-mediated disease¹⁵. When the

multiple “low dose” STZ model (MLDSTZ) of T1D is compared to the NOD model of T1D, there are striking differences in addition to the different gender bias (greater female susceptibility in NOD and other autoimmune models, only males susceptible to MLDSTZ), including the strict Major Histocompatibility Complex (MHC) requirement in NOD model and the ability to adoptively transfer T1D (by NOD lymphocytes) to recipients expressing the diabetogenic H2g7 MHC haplotype, et.al ¹⁶. However, these differences do not significantly change their sensitivity to insulin administration, as the insulin-producing β -cell of pancreas are destructed in both the MLDSTZ and NOD models of T1D. In our study, the induction of insulin expression for restoration of blood glucose levels in T1D mice is a proof-of-concept demonstration of the potential application of the LuminON system for controllable cell therapy, which does not focus on the detailed pathological study of T1D disease. In addition, STZ also has long been used as a tool for creating experimental diabetes because of its relatively specific beta-cell cytotoxic effect. Normally, it is easy and low-cost to construct MLDSTZ mice in a relatively short time (less than two weeks) in laboratory conditions. In comparison, NOD mice are genetically bred and spontaneously develop type 1 diabetes to exhibit high blood glucose levels at 12 weeks of age ¹⁷, which is high-cost and time-consuming. Therefore, we chose MLDSTZ model of T1D to demonstrate the potential applications of LuminON system for precise controllable cell therapy.

3- Using this STZ-low dose model, researchers have found that GLUT-2 was a main target for STZ-toxicity as significant reduction of GLUT-2 protein preceding the onset of hyperglycemia in mice, did the authors thought about measuring GLUT-2 following the infusion of luminGAVPO in mice? Please comment.

Response: We agree with the reviewer’s opinion that GLUT-2 is a main target for STZ-toxicity ¹⁸. However, as referred in question 3, the induction of insulin expression for restoration of blood glucose levels in T1D mice is a proof-of-concept demonstration of the potential application of the LuminON system for controllable cell therapy, which does not focus on the mechanism by which systemic injection of STZ causes β -cell destruction. Thus, measuring blood glucose levels is the most convenient and commonly used method to monitor the progression of T1D. We therefore decided not to measure GLUT-2 in our study, but still greatly appreciate the reviewer’s suggestion.

4- did the authors thought about assessing insulin response to nonglucose secretagogue such arginine?

Response: The reviewer raised a concern to assess insulin response to nonglucose secretagogue. Arginine has long been known to stimulate insulin secretion and has been used as a prototypical agent to represent a nonglucose stimulus of β -cell function ^{19,20}. Typically, arginine has been shown to retain secretagogue activity even when glucose-stimulated insulin secretion is lost. Arginine exerts its insulin-stimulating action by elevating intracellular calcium concentrations as a consequence of its own electrogenic transport into the β -cell, thus effectively bypassing the glycolytic pathway ^{20, 21}. However, in our study, insulin expression and secretion were regulated by the LuminON system, which was controlled by furimazine or blue light and was not related to any secretagogue. We therefore did not assess insulin response to nonglucose secretagogue in our study, but still greatly appreciate the reviewer’s suggestion.

List of new or updated figures and tables.

Fig. 1c, 1e, 1f, 3b, 3d, 4b, 4c, 4d, 4e, 4g, 4i, 4j

Supplementary Fig. 4, 9a, 14, 15, 16, 17, Table 1.

References

1. Jacobs-Tulleneers-Thevissen, D. *et al.* Sustained function of alginate-encapsulated human islet cell implants in the peritoneal cavity of mice leading to a pilot study in a type 1 diabetic patient. *Diabetologia* **56**, 1605-1614 (2013).
2. Jacobs-Tulleneers-Thevissen, D. *et al.* Sustained function of alginate-encapsulated human islet cell implants in the peritoneal cavity of mice leading to a pilot study in a type 1 diabetic patient. *Diabetologia* **56**, 1605-1614 (2013).
3. Vegas, A.J. *et al.* Combinatorial hydrogel library enables identification of materials that mitigate the foreign body response in primates. *Nat Biotechnol* **34**, 345-352 (2016).
4. Vegas, A.J. *et al.* Long-term glycemic control using polymer-encapsulated human stem cell-derived beta cells in immune-competent mice. *Nature medicine* **22**, 306-311 (2016).
5. Soon-Shiong, P. *et al.* Insulin independence in a type 1 diabetic patient after encapsulated islet transplantation. *Lancet* **343**, 950-951 (1994).
6. Tuch, B.E. *et al.* Safety and viability of microencapsulated human islets transplanted into diabetic humans. *Diabetes care* **32**, 1887-1889 (2009).
7. de Groot, M., Schuurs, T.A. & van Schilfgaarde, R. Causes of limited survival of microencapsulated pancreatic islet grafts. *The Journal of surgical research* **121**, 141-150 (2004).
8. Zhang, H., Zhu, S.J., Wang, W., Wei, Y.J. & Hu, S.S. Transplantation of microencapsulated genetically modified xenogeneic cells augments angiogenesis and improves heart function. *Gene therapy* **15**, 40-48 (2008).
9. Wang, D., Tai, P.W.L. & Gao, G. Adeno-associated virus vector as a platform for gene therapy delivery. *Nature reviews. Drug discovery* **18**, 358-378 (2019).
10. Yang, H.K. & Yoon, K.H. Current status of encapsulated islet transplantation. *Journal of diabetes and its complications* **29**, 737-743 (2015).
11. Chen, X., Wang, X., Du, Z., Ma, Z. & Yang, Y. Spatiotemporal control of gene expression in mammalian cells and in mice using the LightOn system. *Curr Protoc Chem Biol* **5**, 111-129 (2013).
12. Wang, X., Chen, X. & Yang, Y. Spatiotemporal control of gene expression by a light-switchable transgene system. *Nature methods* **9**, 266-269 (2012).
13. Mates, L. *et al.* Molecular evolution of a novel hyperactive Sleeping Beauty transposase enables robust stable gene transfer in vertebrates. *Nature genetics* **41**, 753-761 (2009).
14. Aldrich, V.R., Hernandez-Rovira, B.B., Chandwani, A. & Abdulreda, M.H. NOD Mice-Good Model for T1D but Not Without Limitations. *Cell transplantation* **29**, 963689720939127 (2020).
15. Pearson, J.A., Wong, F.S. & Wen, L. The importance of the Non Obese Diabetic (NOD) mouse model in autoimmune diabetes. *Journal of autoimmunity* **66**, 76-88 (2016).
16. Leiter, E.H. & Schile, A. Genetic and Pharmacologic Models for Type 1 Diabetes. *Current protocols in mouse biology* **3**, 9-19 (2013).

17. Leiter, E.H. & Lee, C.H. Mouse models and the genetics of diabetes: is there evidence for genetic overlap between type 1 and type 2 diabetes? *Diabetes* **54 Suppl 2**, S151-158 (2005).
18. Schnedl, W.J., Ferber, S., Johnson, J.H. & Newgard, C.B. STZ transport and cytotoxicity. Specific enhancement in GLUT2-expressing cells. *Diabetes* **43**, 1326-1333 (1994).
19. Floyd, J.C., Jr., Fajans, S.S., Conn, J.W., Knopf, R.F. & Rull, J. Insulin secretion in response to protein ingestion. *The Journal of clinical investigation* **45**, 1479-1486 (1966).
20. Porte, D., Jr. Banting lecture 1990. Beta-cells in type II diabetes mellitus. *Diabetes* **40**, 166-180 (1991).
21. Mao, C.S., Berman, N. & Ipp, E. Loss of entrainment of high-frequency plasma insulin oscillations in type 2 diabetes is likely a glucose-specific beta-cell defect. *American journal of physiology. Endocrinology and metabolism* **287**, E50-54 (2004).

Reviewers' Comments:

Reviewer #1:

Remarks to the Author:

The authors have addressed my comments.

Reviewer #2:

Remarks to the Author:

The authors addressed the concerns raised and responded constructively to suggestions. Congratulations to a very nice body of work.

Thank you,

Reviewer #3:

Remarks to the Author:

In their revised manuscript, Ting Li et al have addressed some of the reviewers' comments which have strengthened in part their article. I will recommend their article to be accepted for publication after addressing the following comments:

- 1- Fig1 , panels C & f: the legend is too much attached to the upper limit of the bargraph, please fix it. Also please put unit of measures to the horizontal legend.
- 2- Please fix the upper limit of Y axis in panel f and same comment for Fig 2
- 3- Please add missing p values to fig panels when applicable
- 4- Make sure that all acronyms that are cited all over the manuscript and in the figures legend are spelled when firstly cited.
- 5- More details regarding the statistical test are required , is a corrected applied ?please check all over the manuscript and correct when applicable.
- 6- For fig 3: add missing p val to panels g, f. Remove red underline on DIC in panel d.
- 7- Same comments for supplementary figures regarding the p val.
- 8- Supplemental fig 1 &2 can be consolidated in one figure.
- 9- Same comment as above: suppl fig 4, and 6 can be consolidated in one figure
- 10- Suppl fig 5 panel: please add a relative quantif to this panel.
- 11- Sup fig 5, panel c: please fix the limit of Y axis.
- 12- Sup fig 6: please add missing p val to panels.
- 13- Sup fig 67: please add p val to panels if applicable. fix the limit of Y axis in panel b.
- 14- Supp fig 11: add missing p val and fix the y axis
- 15- supp fig 13: panel a is not clear enough please provide a legend to A->L and to 01->24
- 16- supp fig 13: add missing p val and fix the y axis of panel
- 17- supp fig 14: fix the y axis of panel
- 18- supp fig 16 & 17: while H&E is shown or representative pic is shown there's no clear or evident difference that could appreciated between the different groups, so a quantification of the cells infiltrate or lesions will be appreciated
- 19- supp fig 16 : also changes in the ratios weight of delineated organs (spleens, lung kidneys hearts) per total body weight in all groups rather showing the changes in total body weight will be further informative

Response to the reviewers

Reviewer #1 (Remarks to the Author):

The authors have addressed my comments.

We appreciate your positive comments as well as the highly constructive suggestions.

Reviewer #2 (Remarks to the Author):

The authors addressed the concerns raised and responded constructively to suggestions.
Congratulations to a very nice body of work.

Thank you,

We appreciate your positive comments as well as the highly constructive suggestions.

Reviewer #3 (Remarks to the Author):

In their revised manuscript, Ting Li et al have addressed some of the reviewers' comments which have strengthened in part their article. I will recommend their article to be accepted for publication after addressing the following comments:

We appreciate your positive comments as well as the highly constructive suggestions.

1- Fig1, panels C & f: the legend is too much attached to the upper limit of the bargraph, please fix it. Also please put unit of measures to the horizontal legend.

Response: We have fixed the legend of the barograph in Fig. 1c and 1f, and have added the unit of measures to the horizontal legend in the revised manuscript.

2- Please fix the upper limit of Y axis in panel f and same comment for Fig 2

Response: We have fixed the upper limit of Y axis in panel f and Fig. 2.

3- Please add missing p values to fig panels when applicable

Response: We have confirmed that all p values to figure panels are added when applicable.

4- Make sure that all acronyms that are cited all over the manuscript and in the figures, legend are spelled when firstly cited.

Response: We have confirmed that all acronyms cited all over the manuscript have been spelled when firstly cited.

5- More details regarding the statistical test are required, is a corrected applied? please check all over the manuscript and correct when applicable.

Response: We have checked the statistical test all over the manuscript.

6- For fig 3: add missing p val to panels g, f. Remove red underline on DIC in panel d.

Response: We have added p values to Fig. 3g and 3f and have removed red underline on DIC in Fig. 3d.

7- Same comments for supplementary figures regarding the p val.

Response: We have added the p values in supplementary figures.

8- Supplemental fig 1 &2 can be consolidated in one figure.

Response: We have consolidated Supplementary Fig. 1 and Supplementary Fig. 2 in one figure following the reviewer's suggestion.

9- Same comment as above: suppl fig 4, and 6 can be consolidated in one figure

Response: We have consolidated Supplementary Fig. 4 and Supplementary Fig. 6 in one figure following the reviewer's suggestion.

10- Suppl fig 5 panel: please add a relative quantify to this panel.

Response: We have added a relative quantify to Supplementary Fig. 5.

11- Sup fig 5, panel c: please fix the limit of Y axis.

Response: We have fixed the limit of Y axis in Supplementary Fig. 5c.

12- Sup fig 6: please add missing p val to panels.

Response: We have added p values to Supplementary Fig. 6.

13- Sup fig 9: please add p val to panels if applicable. fix the limit of Y axis in panel b.

Response: We have added p values to the Supplementary Fig. 9 and fixed the limit of Y axis in Supplementary Fig. 9b.

14- Supp fig 11: add missing p val and fix the y axis

Response: We have added p values and fixed the limit of Y axis in Supplementary Fig. 11.

15- supp fig 13: panel a is not clear enough please provide a legend to A->L and to 01->24

Response: We have provided a legend to A->L and to 01->24 in the revised manuscript.

16- supp fig 13: add missing p val and fix the y axis of panel

Response: We have added p values and fixed the limit of Y axis in Supplementary Fig. 13.

17- supp fig 14: fix the y axis of panel

Response: We have fixed the limit of Y axis in Supplementary fig 14.

18- supp fig 16 & 17: while H&E is shown or representative pic is shown there's no clear or evident difference that could appreciated between the different groups, so a quantification of the cells infiltrate or lesions will be appreciated

Response: We thank the reviewer for the valuable suggestion. We have provided the quantification results of the H&E staining (**Supplementary Fig. 14b, e and h**), and we also used arrows to show the alveolar stenosis in lung and granulocyte infiltration in kidney in the H&E stained pictures in the revised manuscript (**Supplementary Fig. 14h**).

19- supp fig 16: also changes in the ratios weight of delineated organs (spleens, lung kidneys hearts) per total body weight in all groups rather showing the changes in total body weight will be further informative

Response: We agree with the reviewer's opinion that the ratios weight of delineated organs per total body weight may provide more information for the toxicity of the chemical to the animals. However, our current data including body weight and organ sections is a universal way to show the toxicity of an agent to the animals¹⁻⁸. These data showed that oral administration of furimazine has little toxicity to the animals, which has addressed the reviewer's concern about the toxicity of furimazine. Further investigation of the specific details of toxicity may enable the study to appear to be out of focus.

Therefore, we decided not to show the ratios weight of delineated organs per total body weight in this study, but still greatly appreciate the reviewer's valuable suggestion.

List of new or updated Figures or Tables.

Fig. 1c, 1f, 2b, 2c, 2f, 2h, 2i, 2k, 2l, 2m, 2n, 3d, 3f, 3g, 4c, 4d, 4e,
Supplementary Fig. 1e, 1f, 1g, 1h, 1i, 3, 4a, 4b, 7, 9, 11b, 12, 13, 14, 15
Supplementary Table 1

Reference

1. Minzel, W. *et al.* Small Molecules Co-targeting CKIalpha and the Transcriptional Kinases CDK7/9 Control AML in Preclinical Models. *Cell* **175**, 171-185 e125 (2018).
2. Du, H. *et al.* Antitumor Responses in the Absence of Toxicity in Solid Tumors by Targeting B7-H3 via Chimeric Antigen Receptor T Cells. *Cancer Cell* **35**, 221-237 e228 (2019).
3. Aupy, P. *et al.* Identifying and Avoiding tcDNA-ASO Sequence-Specific Toxicity for the Development of DMD Exon 51 Skipping Therapy. *Mol Ther Nucleic Acids* **19**, 371-383 (2020).
4. Liu, L. *et al.* Triose Kinase Controls the Lipogenic Potential of Fructose and Dietary Tolerance. *Cell Metab* **32**, 605-618 e607 (2020).
5. Zhao, Y. *et al.* SoNar, a Highly Responsive NAD⁺/NADH Sensor, Allows High-Throughput Metabolic Screening of Anti-tumor Agents. *Cell metabolism* **21**, 777-789 (2015).
6. Gu, L. *et al.* Discovery of Dual Inhibitors of MDM2 and XIAP for Cancer Treatment. *Cancer Cell* **30**, 623-636 (2016).
7. Lee, H. *et al.* Recognition of Semaphorin Proteins by *P. sordellii* Lethal Toxin Reveals Principles of Receptor Specificity in Clostridial Toxins. *Cell* **182**, 345-356 e316 (2020).
8. Velkov, T. *et al.* Structure, Function, and Biosynthetic Origin of Octapeptin Antibiotics Active against Extensively Drug-Resistant Gram-Negative Bacteria. *Cell Chem Biol* **25**, 380-391 e385 (2018).